# RobustTSF: Towards Theory and Design of Robust Time Series Forecasting with Anomalies

**Hao Cheng[1], Qingsong Wen[2]\*, Yang Liu[1], and Liang Sun[2]**
[1]University of California, Santa Cruz, [2]Alibaba Group
{haocheng, yangliu}@ucsc.edu,
qingsongedu@gmail.com, liang.sun@alibaba-inc.com

## Abstract

Time series forecasting is an important and forefront task in many real-world applications. However, most of time series forecasting techniques assume that the training data is clean without anomalies. This assumption is unrealistic since the collected time series data can be contaminated in practice. The forecasting model will be inferior if it is directly trained by time series with anomalies. Thus it is essential to develop methods to automatically learn a robust forecasting model from the contaminated data. In this paper, we first statistically define three types of anomalies, then theoretically and experimentally analyze the *loss robustness* and *sample robustness* when these anomalies exist. Based on our analyses, we propose a simple and efficient algorithm to learn a robust forecasting model. Extensive experiments show that our method is highly robust and outperforms all existing approaches. The code is available at `https://github.com/haochenglouis/RobustTSF`.

## 1 Introduction

As the development of Internet of Things, the monitoring of cloud computing, and the increase of connected sensors, the applications of time series data increase significantly (Lai et al., 2021; Wen et al., 2022; Lim & Zohren, 2021). However, many time series data collected are contaminated by noises and anomalies. In time series forecasting with anomalies (TSFA), we study how to make robust and accurate forecasting on contaminated time series data. The earliest work of TSFA can date back to (Connor et al., 1994), where the classic **detection-imputation-retraining** pipeline is proposed. In the pipeline, a forecasting model is first trained using the data with outliers and noises. Then the trained model is used to predict values for each time step (**detection**). If the predicted value is far from the observed value, we will regard this time step as anomaly time step and then use the predicted value to replace the observed value on this time step (**imputation**). After imputation, the imputed (filtered) data are utilized to **retrain** a new forecasting model. The detection-imputation-retraining pipeline can work with any forecasting algorithm and improves performance in many TSFA tasks. However, it is very sensitive to the training threshold and anomaly types. Note that TSFA is also related to time series anomaly detection (TSAD) (Blázquez-García et al., 2021), especially a forecasting model is learned to compute the reconstruction error, which is then used to determine anomaly.

"TSFA shares similarities with learning with noisy labels (LNL) (e.g., see (Natarajan et al., 2013; Song et al., 2022)), but their settings differ significantly. LNL typically involves datasets with clean input samples $X$ and noisy labels $\tilde{Y}$, such as mislabeled images in image classification tasks. In time series forecasting, we typically use past history, sometimes aided by external factors, to predict future targets. In this context, we denote all covariates as $X$ and the target we are predicting as $Y$. Notably, TSFA faces the challenge of handling anomalies in both covariates and targets simultaneously. This characteristic makes applying LNL methods directly to TSFA challenging. Moreover, the diverse nature of time series data in real-world scenarios leads to various types of anomalies. Existing algorithms struggle to handle these diverse anomalies effectively. For instance, (Li et al., 2022) addresses a similar problem to TSFA, emphasizing anomaly detection and drawing from LNL ideas. However, the small loss criterion in LNL does not universally apply to common anomaly types in TSFA, resulting in inconsistent improvements across TSFA anomaly types."

---

\*Corresponding author

In this paper, we introduce a unified framework for TSFA, bridging the gap between TSFA and LNL. We address various anomaly types commonly found in real-world applications. Within this framework, we present RobustTSF, an efficient algorithm that outperforms state-of-the-art methods in both single-step and multi-step forecasting. In contrast to the detection-imputation-retraining pipeline, RobustTSF identifies informative samples by assessing the variance between the original input time series and its trend. Subsequently, it employs a robust loss function to improve the forecasting process. This distinctive approach is substantiated by our analyses comparing TSFA and LNL. Contributions can be summarized as follows:

- We analyze the impact of three formally defined anomalies on model performance, exploring *loss robustness* with target anomalies and *sample robustness* with covariate anomalies.
- We propose RobustTSF, an efficient algorithm for TSFA, which is both theoretically grounded and empirically superior to existing TSFA approaches.
- We conduct extensive experiments, including single-step and multi-step time series forecasting with different model structures, to validate our method's performance.
- To the best of our knowledge, this is the first study to extend the theory of LNL to time series forecasting, which builds a bridge between LNL and TSFA tasks.

## 2 RELATED WORKS

**Robust Time Series Forecasting:** There is a line of works to learn a robust forecasting model in different data bias settings: (Yoon et al., 2022) proposes to make forecasting model robust to input perturbations, motivated by adversarial examples in classification (Goodfellow et al., 2015). (Arik et al., 2022; Woo et al., 2022; Wang et al., 2021a) propose to make the forecasting model robust to non-stationary time series under concept drift or distribution shift assumption (Ditzler et al., 2015; Lu et al., 2018). Different from these settings, our paper considers learning a robust time series forecasting model under the existence of anomalies (TSFA). Some existing approaches (Connor et al., 1994; Bohlke-Schneider et al., 2020; Li et al., 2022) deploy detection-imputation-retraining pipeline, or use small loss criterion to TSFA tasks which lack a deep understanding of how these anomalies affect model performance which makes their methods less explainable.

**Robust Regression** Our setting is also similar to robust regression (Pensia et al., 2020; Kong et al., 2022), which has some rigorous guarantees when the input and the label have anomalies. However, the theoretical guarantees are based on (generalized) *linear* regression. The alternating minimization (Pensia et al., 2020) also needs to calculate the inverse of $X^T X$. For time series forecasting with deep neural networks (DNNs), it is not practical to calculate the inverse of the source data since the model of DNNs is not the simple linear layer. Thus it is hard to generalize the algorithm and analyses in robust linear regression into robust time series forecasting with DNNs.

**Learning with Noisy Labels:** The goal of LNL is to mitigate the noisy label effect in classification and learn a robust model from the noisy dataset (Natarajan et al., 2013; Liu, 2021). The problem of LNL has been researched for a long time in the machine learning community, and many methods have been proposed in recent years, such as sample selection (Jiang et al., 2018; Han et al., 2018), robust loss design (Zhang & Sabuncu, 2018; Liu & Guo, 2020; Cheng et al., 2021a;b; Zhu et al., 2021a), transition matrix estimation (Patrini et al., 2017; Zhu et al., 2021b), etc. Among all these methods, arguably the most efficient treatment is to adopt robust losses, since sample selection and noise transition matrix estimation always involve training multiple networks or need multi-stage training. We say a loss $\ell$ is **noise-robust** in LNL if it satisfies the following equation (Ghosh et al., 2017; Xu et al., 2019; Ma et al., 2020; Liu & Guo, 2020):

$$\arg\min_{f \in \mathcal{F}} \mathbb{E}_{(X,\widetilde{Y})}[\ell(f(X), \widetilde{Y}] = \arg\min_{f \in \mathcal{F}} \mathbb{E}_{(X,Y)}[\ell(f(X), Y)], \tag{1}$$

where $f$ is the classifier, $X$ is the input, $Y$ and $\widetilde{Y}$ are clean labels and noisy labels, respectively. Equation (1) implies that minimizing $\ell$ over a clean dataset with respect to $f$ is equivalent to minimizing $\ell$ over a noisy dataset, demonstrating $\ell$'s robustness to label noise. However, applying robust losses from LNL directly to TSFA faces two challenges: 1) TSFA deals with regression problems rather than classification problems where label noise is typically modeled. The effectiveness of robust losses from LNL in TSFA is uncertain without proper anomaly modeling. 2) In LNL, noise is limited to $Y$, while TSFA involves anomalies in both $X$ and $Y$. We address these issues by analyzing loss and sample robustness in Sections 4 and 5 of this paper.

## 3 PRELIMINARY AND FORMULATION

**Problem Formulation:** Let $(z_1, z_2, \cdots, z_T)$ be a time series (without anomalies) of length $T$. Assume that the time series is partitioned into a set of time windows with size $\alpha$ $(1 < \alpha < T)$, where each window is a sequence of predictors $\boldsymbol{x}_n = (z_n, \cdots, z_{n+\alpha-1})$ and its corresponding label is $y_{\boldsymbol{x}_n} = z_{n+\alpha}$, where $n \in \{1, 2, \cdots, T - \alpha\}$. This partition gives the training set $D = \{(\boldsymbol{x}_1, y_{\boldsymbol{x}_1}), \cdots, (\boldsymbol{x}_N, y_{\boldsymbol{x}_N})\}$ where $N = T - \alpha$, which can be assumed to be drawn according to an unknown distribution, $\mathcal{D}$, over $\mathcal{X} \times \mathcal{Y}$. In real-world applications, time series may have anomalies (anomaly types are defined later), and we can only observe time series with anomaly signals. The observed training set is denoted by $\widetilde{D} = \{(\widetilde{\boldsymbol{x}}_1, \widetilde{y}_{\boldsymbol{x}_1}), \cdots, (\widetilde{\boldsymbol{x}}_N, \widetilde{y}_{\boldsymbol{x}_N})\}$, where ~ denotes the observed value. The robust forecasting task aims to identify a model $f$ that maps $X$ to $Y$ accurately using the observed training set $\widetilde{D}$ with anomalies.

**Anomaly types:** In the paper, we mainly consider point-wise anomalies (we also provide preliminary studies for subsequence anomaly in Section 7.3). Define the noise (anomaly) rate as $\eta$ where $0 < \eta < 1$. The observed value $\widetilde{z}_t$ in the time series can be represented as:

$$\widetilde{z}_t = \begin{cases} z_t & \text{with probability } 1 - \eta \\ z_t^{\mathrm{A}}, & \text{with probability } \eta \end{cases},$$

where $z_t$ is the ground-true value and $z_t^{\mathrm{A}}$ is the value of anomaly. We consider three types of $z_t^{\mathrm{A}}$ as

- **Constant type anomaly**: $z_t^{\mathrm{A}} = z_t + \epsilon$, $\epsilon$ is constant.
- **Missing type anomaly**: $z_t^{\mathrm{A}} = \epsilon$, $\epsilon$ is constant.
- **Gaussian type anomaly**: $z_t^{\mathrm{A}} = z_t + \epsilon, \epsilon \sim \mathcal{N}(0, \sigma^2)$.

The $\mathcal{N}$ denotes Gaussian distribution, and the $\epsilon$ is noise scale. Note that $\epsilon$ in Constant type anomaly and $\sigma^2$ in Gaussian type anomaly should be large so that they can represent anomalies instead of small random noise (e.g., white noise). Generally, the noise scale is within the range of $(-\infty, +\infty)$.

Constant anomalies exhibit a fixed difference between the anomaly value and the ground-truth value, Missing anomalies are independent of the ground-truth value, and Gaussian anomalies follow a Gaussian distribution. While (Connor et al., 1994) used Constant and Gaussian anomaly types for experiments without explicit definitions, they effectively simulate real-world anomalies. For instance, Missing anomalies represent missing sensor data ($\epsilon = 0$), Constant and Gaussian anomalies model sensor functionality affected by external factors like temperature or humidity. We provide theoretical and experimental analyses of these anomaly types in this paper, with illustrations in Appendix B.

## 4 LOSS ROBUSTNESS: ANOMALY EFFECT IN $Y$

Understanding how anomalies affect model performance is hard when anomalies exist in both $X$ and $Y$. Thus we first assume the input $X$ is clean and anomalies only exist in $Y$. In this case, $\widetilde{D} = \{(\boldsymbol{x}_1, \widetilde{y}_{\boldsymbol{x}_1}), \cdots, (\boldsymbol{x}_N, \widetilde{y}_{\boldsymbol{x}_N})\}$ (we can assume these data are non-overlapping segments of the time series). $\widetilde{y}_{\boldsymbol{x}_n}$ can be represented as:

$$\widetilde{y}_{\boldsymbol{x}_n} = \begin{cases} y_{\boldsymbol{x}_n} & \text{with probability } 1 - \eta \\ y_{\boldsymbol{x}_n}^A, & \text{with probability } \eta \end{cases}.$$

Thus this setting is similar to the vanilla LNL setting except that for time series forecasting, the problem is regression instead of classification. Next, we examine the robustness of loss functions for different anomaly types defined earlier.

**Theorem 1.** *Let $\ell$ be the loss function and $f$ be the forecasting model. Under Constant and Missing type anomalies with anomaly rate $\eta < 0.5$, if for each $\boldsymbol{x}$, $\ell(f(\boldsymbol{x}), y_{\boldsymbol{x}}) + \ell(f(\boldsymbol{x}), y_{\boldsymbol{x}}^A) = C_{\boldsymbol{x}}$, where $C_{\boldsymbol{x}}$ is constant respect to the choice of $f$. Then we have:*

$$\mathbb{E}_{\widetilde{\mathcal{D}}}[\ell(f(X), \widetilde{Y})] = \gamma_1 \mathbb{E}_{\mathcal{D}}[\ell(f(X), Y)] + \gamma_2, \tag{2}$$

*where $\gamma_1 > 0$ and $\gamma_2$ are constants respect to $f$.*

Our proposed Theorem 1 suggests that if a loss function satisfies $\ell(f(\boldsymbol{x}), y_{\boldsymbol{x}}) + \ell(f(\boldsymbol{x}), y_{\boldsymbol{x}}^A) = C_{\boldsymbol{x}}$, then it is robust to constant and missing type anomalies. Equation (2) implies that minimizing $\ell$ under $\widetilde{\mathcal{D}}$ with respect to $f$ is identical to minimizing $\ell$ under $\mathcal{D}$ (Equation (1)).

Figure 1: Visualization of time series with Gaussian anomalies in different positions. (a): Clean input time series which is the sine function of time steps. We normalize the time series to 0 mean and 1 std. (b) (c) (d): Time series (sine) with Gaussian anomalies located in front, middle and back of the input time series, respectively.

**Proposition 1.** *From Theorem 1, MAE is robust to Constant and Missing type anomalies while MSE is not robust.*

**Proposition 2.** *Assume $C_{\boldsymbol{x}}$ can be estimated correctly. Let $\ell$ be any possible loss function operated on $\widehat{f}(\boldsymbol{x})$ and $\widetilde{y}_{\boldsymbol{x}}$. Define $\ell_{norm} = \frac{\ell}{C_{\boldsymbol{x}}}$. Then $\ell_{norm}$ is robust to Constant and Missing type anomalies.*

Proposition 1 analyzes the robustness of common MAE and MSE loss functions. Interestingly, MAE has been found to be robust to symmetric label noise in classification as well (Ghosh et al., 2017). However, the robustness conditions in (Ghosh et al., 2017) and our work differ. In their work, robustness requires $C_{\boldsymbol{x}} \equiv C$, whereas our approach does not necessitate such a strong condition. While Theorem 1 addresses the scenario in which $C_x$ remains constant with respect to the forecasting model, Proposition 2 explores the case where $C_x$ is not constant. In this situation, any loss function can be made robust to Constant and Missing type anomalies, provided that accurate estimation of $C_{\boldsymbol{x}}$ is achievable. However, practical estimation of $C_{\boldsymbol{x}}$ may be challenging, particularly when we lack prior knowledge of the anomaly generation process. In summary, Theorem 1 represents a generalization from noisy labels in classification to anomalies in regression.

**Theorem 2.** *Let $\ell$ be MAE or MSE loss function and $f$ be the forecasting model. Under Gaussian type anomaly, let $f^* = \arg\min_{f \in \mathcal{F}} \mathbb{E}_{\widetilde{\mathcal{D}}}[\ell(f(X), \widetilde{Y})]$. Then we have $f^*(X) = Y$.*

Theorem 2 implies that MAE and MSE are both robust to Gaussian anomalies in $Y$. Intuitively, this is understandable since the mean of $y_{\boldsymbol{x}_n}^A$ is the ground truth $y_{\boldsymbol{x}_n}$. It is also worth noting that these conclusions are made statistically. If the size of the training set is relatively small or the anomaly rate is high, the performance may still drop. However, in our experiments, we find that MAE can be very robust in some real-world datasets.

Detailed proofs for the above Theorems and Propositions can be found in Appendix A. It may appear straightforward that MAE is more robust than MSE, but the extent and conditions of MAE's robustness are not well understood. For instance, prior research in label noise (LNL) (Ghosh et al., 2017) has demonstrated MAE's inherent robustness to symmetric label noise and, under certain conditions, to asymmetric label noise in classification tasks. However, these analyses are classification-focused. To our knowledge, no study has explored how MAE performs in regression when dealing with various types of anomalies in time series forecasting. Notably, previous works (Connor et al., 1994; Bohlke-Schneider et al., 2020; Li et al., 2022) predominantly use MSE in TSFA tasks. Our analysis indicates that MAE is inherently robust to Constant and Missing type anomalies, while both MAE and MSE exhibit robustness to Gaussian type anomalies. Experimental evidence supporting these theoretical findings is presented in Appendix C.

## 5 SAMPLE ROBUSTNESS: ANOMALY EFFECT IN $X$

In this section, we analyze how anomalies in $X$ impact performance while keeping the labels clean. It's worth noting that a related study (Li et al., 2021) also investigates the influence of input time series outliers on prediction performance, using a point-wise anomaly model (Equation (1) in (Li et al., 2021)). In particular, (Li et al., 2021) introduces a metric called SIF to quantify outlier effects on predictions. However, calculating SIF necessitates knowledge of the outlier generation process and does not propose an efficient algorithm for mitigating outlier effects. In our context, we lack information about outlier (anomaly) generation, mirroring real-world scenarios where only observed time series are available. Our goal is to leverage an understanding of anomaly effects in $X$ to develop an efficient performance improvement algorithm. We begin by assuming that *not all input samples equally contribute to performance, even if they statistically exhibit the same number of anomalies.*

Our assumption highlights the significance of anomaly position. To validate this, we conducted simulations involving time series with anomalies placed at various positions. For instance, Figure 1

Table 1: Evaluating how different positions of anomalies in the input time series affect prediction performance. For each dataset, we simulate different types of anomalies with anomaly rates 0.1 and 0.2. The loss is MAE and the network structure is LSTM. Detailed setting is left in Appendix E.

| Anomaly Type | Electricity with Anomaly Position | | | Traffic with Anomaly Position | | |
|---|---|---|---|---|---|---|
| | front | middle | back | front | middle | back |
| Clean | 0.182/0.070 | 0.182/0.070 | 0.182/0.070 | 0.194/0.105 | 0.194/0.105 | 0.194/0.105 |
| Const. ($\eta = 0.1$) | 0.182/0.070 | 0.183/0.071 | 0.195/0.078 | 0.196/0.105 | 0.200/0.115 | 0.215/0.131 |
| Const. ($\eta = 0.2$) | 0.182/0.069 | 0.183/0.071 | 0.219/0.094 | 0.194/0.109 | 0.195/0.106 | 0.236/0.153 |
| Missing. ($\eta = 0.1$) | 0.182/0/070 | 0.186/0.073 | 0.204/0.083 | 0.203/0.118 | 0.203/0.120 | 0.221/0.127 |
| Missing. ($\eta = 0.2$) | 0.182/0.070 | 0.183/0.071 | 0.223/0.105 | 0.204/0.116 | 0.205/0.116 | 0.251/0.160 |
| Gaussian. ($\eta = 0.1$) | 0.184/0.071 | 0.187/0.071 | 0.196/0.075 | 0.204/0.120 | 0.206/0.114 | 0.211/0.133 |
| Gaussian. ($\eta = 0.2$) | 0.184/0.171 | 0.186/0.072 | 0.213/0.087 | 0.206/0.121 | 0.212/0.127 | 0.226/0.147 |

(b), (c), and (d) depict a simple example involving Gaussian type anomalies, where the number of anomalies remains consistent. Our experiments, conducted on the Electricity and Traffic datasets, demonstrate the impact of anomaly position on model performance, as summarized in Table 1. Notably, when anomalies are located at the front or middle of input time series, performance remains close to that of clean data training, with minimal performance degradation. Conversely, when anomalies are positioned at the end of time series, performance significantly declines as the anomaly rate increases. This is understandable since the label is situated towards the end of the input time series. Further evidence on various time series datasets can be found in Appendix D. Interestingly, similar findings have been reported in temporal graph learning (Wang et al., 2021b), which emphasizes the informativeness of more recent edges for target predictions."

## 6  INSIGHTS AND OUR ALGORITHM

In this part, based on the analyses in Section 4 and Section 5, we show how to achieve loss robustness in TSFA as Equation (1) in LNL when anomalies exist in both inputs and labels.

### 6.1  CONNECTION BETWEEN LNL AND TSFA

Our design proceeds as follows:

• Denote the original observed inputs and labels as $\widetilde{X}$ and $\widetilde{Y}$. We first select $X^{'}$ from $\widetilde{X}$ where the anomalies of $X^{'}$ are sparse in the time series points located near the label. Denote the ground-truth label and observed label of $X^{'}$ as $Y^{'}$ and $\widetilde{Y'}$, respectively. From our analyses of sample robustness in Section 5, training on $(X^{'}, Y^{'})$ has very similar prediction performance compared to training on $(X,Y)$. *I.e.,* $X^{'}$ acts like clean input time series. Mathematically, we have

$$\arg\min_f \mathbb{E}_{X,Y}[l(f(X), Y)] \approx \arg\min_f \mathbb{E}_{X^{'},Y^{'}}[l(f(X^{'}), Y^{'})].$$

• Since the selection process does not involve the labels, the distribution of $\widetilde{Y'}$ with respect to $X^{'}$ is unchanged. Let $\ell$ be the robust loss (MAE) in the setting of anomalies. From our analyses of loss robustness in Section 4, we have:

$$\arg\min_f \mathbb{E}_{X^{'},\widetilde{Y'}}[l(f(X^{'}), \widetilde{Y'})] \approx \arg\min_f \mathbb{E}_{X^{'},Y^{'}}[l(f(X^{'}), Y^{'})].$$

• Combining the above two equations, we have:

$$\arg\min_f \mathbb{E}_{X^{'},\widetilde{Y'}}[l(f(X^{'}), \widetilde{Y'})] \approx \arg\min_f \mathbb{E}_{X^{'},Y^{'}}[l(f(X^{'}), Y^{'})] \approx \arg\min_f \mathbb{E}_{X,Y}[l(f(X), Y)]. \quad (3)$$

Equation (3) implies that by using selective samples to train the model, the performance can be close to the clean samples training without anomalies in both inputs and labels. Equation (3) can be viewed as a generalization of Equation (1) which considers the noise in the inputs in TSFA tasks.

We use $\approx$ instead of $=$ because our input time series contain anomalies. In the absence of anomalies in the input, as guaranteed by Section 4, strict equality in Equation (3) can be achieved. Unlike LNL, which assumes clean inputs, time series data can be contaminated. Fortunately, our analysis in Section 5 shows that by selecting input time series with minimal anomalies near the label, we can make them comparable to clean series, as shown in Table 1. The use of $\approx$ reflects this observation. In summary, while LNL relies on strict robustness, our approach incorporates a selection procedure to ensure robustness in the presence of potential anomalies.

---

**Algorithm 1** RobustTSF Algorithm

---

**Input:** Initialized DNN, Time series $\widetilde{\boldsymbol{z}} = \{\widetilde{z}_1, \cdots, \widetilde{z}_T\}$, Robust loss $\ell$, Number of epochs $E$, Batch size $M$.
**Iteration:**
    1. Input the time series $\widetilde{\boldsymbol{z}}$ into Equation (4) to get the trend of time series $\boldsymbol{s}$.
    2. Segment $\widetilde{\boldsymbol{z}}$ and $\boldsymbol{s}$ to build the dataset $\widetilde{D} = \{(\widetilde{\boldsymbol{x}}_1, \boldsymbol{s}_1, \widetilde{y}_{\boldsymbol{x}_1}), \cdots, (\widetilde{\boldsymbol{x}}_N, \boldsymbol{s}_N, \widetilde{y}_{\boldsymbol{x}_N})\}$
    **for** $e = 1$ to $E$ **do**
        **for** $m = 1$ to $\frac{N}{M}$ **do**
            3. Randomly select $M$ triplets from $\widetilde{D}$
            4. Calculate anomaly score for each triplet from Equation (5).
            5. Train DNN on triplets using robust loss $\ell$ from Equation (6).
        **end for**
    **end for**
**Output:** DNN model

---

## 6.2 ROBUSTTSF ALGORITHM

From Section 6.1, to achieve robustness in TSFA (Equation (3)), it suffices to show how to select $X'$ from $X$. We derive an efficient algorithm as follows:

Given a time series $\widetilde{\boldsymbol{z}} = \{\widetilde{z}_1, \cdots, \widetilde{z}_T\}$, we first calculate its trend $\boldsymbol{s} = \{s_1, \cdots, s_T\}$ by solving:

$$\min_{\boldsymbol{s}} \sum_{t=1}^{T} |\widetilde{z}_t - s_t| + \lambda \cdot \sum_{t=2}^{T-1} |s_{t-1} - 2 \cdot s_t + s_{t+1}|, \tag{4}$$

where $\lambda > 0$ is the hyper-parameter, $s_t$ is the $t$-th value in $\boldsymbol{s}$. It is worth noting that Equation (4) is a variation of the original trend filtering algorithm (Kim et al., 2009). We change $(\widetilde{z}_t - s_t)^2$ to $|\widetilde{z}_t - s_t|$ to improve robustness. After trend filtering, we can segment $\boldsymbol{z}$ and $\boldsymbol{s}$ to get the training set $\widetilde{D} = \{(\widetilde{\boldsymbol{x}}_1, \boldsymbol{s}_1, \widetilde{y}_{\boldsymbol{x}_1}), \cdots, (\widetilde{\boldsymbol{x}}_N, \boldsymbol{s}_N, \widetilde{y}_{\boldsymbol{x}_N})\}$ which consists of $N$ triplets. Note that $\boldsymbol{s}_n$ is the trend of $\widetilde{\boldsymbol{x}}_n$. Let $A(\widetilde{\boldsymbol{x}}_n)$ denote the anomaly score of $(\widetilde{\boldsymbol{x}}_n, \boldsymbol{s}_n, \widetilde{y}_{\boldsymbol{x}_n})$, which is defined as follows:

$$A(\widetilde{\boldsymbol{x}}_n) = \sum_{k=1}^{K} w(k) \cdot |\widetilde{\boldsymbol{x}}_n^k - \boldsymbol{s}_n^k|, \tag{5}$$

where $K$ is the length of the input $\widetilde{\boldsymbol{x}}_n$, $\widetilde{\boldsymbol{x}}_n^k$ and $\boldsymbol{s}_n^k$ are the $k$-th value in $\widetilde{\boldsymbol{x}}_n$ and $\boldsymbol{s}_n$, respectively. $|\widetilde{\boldsymbol{x}}_n^k - \boldsymbol{s}_n^k|$ denotes the extent of anomaly at time step $k$. $w(k)$ is a weighting function that is designed non-decreasing, and we consider the following two functions for $w(k)$:

- Exponential: $w(k) = e^{-(k-K)^2}$;    Dirac: $w(k) = 0$ if $k < K'$, $w(k) = 1$ if $K' \leq k < K$.

The reason for designing $w(k)$ as a non-decreasing function is because the position of anomalies matters for the prediction performance, as we analyzed in Section 5. we enlarge the weights when anomalies are close to the labels. In practice, we find Dirac function performs slightly better than exponential functions. We design the final loss in RobustTSF as:

$$L = \sum_{n=1}^{N} \mathbb{1}(A(\widetilde{\boldsymbol{x}}_n) < \tau) \cdot l(\widetilde{\boldsymbol{x}}_n, \widetilde{y}_{\boldsymbol{x}_n}), \tag{6}$$

where $\mathbb{1}()$ represents the indicator function, equaling 1 when the condition is met and 0 otherwise. $\tau > 0$ serves as the threshold and $l$ is selected as the MAE loss function. It is worth noting that our primary focus is on discerning the extent of anomalies within time series points situated in proximity to the label, followed by appropriate filtering, rather than merely determining whether a given time series point qualifies as an anomaly. This focus steers our use of the hyper-parameter $\tau$ in Equation (6), streamlining and simplifying the anomaly detection within our method. Experiments in the Appendix F.6 demonstrate that setting $\tau$ to 0.3 yields favorable results across diverse datasets and settings. We term our approach RobustTSF, summarized in Algorithm 1. There are two major differences between our approach and previous detection-imputation-retraining pipeline (Connor et al., 1994; Bohlke-Schneider et al., 2020):

- Our method does not need a pre-trained forecasting DNN to detect anomalies. Instead, our method only needs to calculate anomaly score (see Equation (4) and (5)), which is more time efficient.

- In the detection-imputation-retraining pipeline, removing anomalies post-detection is a viable approach but can introduce time series discontinuities, posing a crucial challenge in TSFA. (Connor et al., 1994) employs offline value imputation, while (Bohlke-Schneider et al., 2020) opts for online imputation. However, these imputations can be noisy and lack robust theoretical foundations, which RobusTSF mitigates by eliminating the need for imputation. Implicit detection in our approach serves efficient sample selection. With sample selection and robust loss, we achieve the desired robustness, as expressed in Equation (3).

While there are potential alternatives to replace Equation (4) for trend calculation and selective approaches to replace Equation (5), our primary focus in this paper is to establish a connection between LNL and TSFA. Our aim is to demonstrate that we can leverage insights from LNL to significantly enhance TSFA performance. Therefore, we've opted to keep our method simple and effective, leaving room for future improvements and exploration of alternative approaches.

## 7 EXPERIMENTS

Here we demonstrate the effects of RobusTSF on TSFA tasks. We simply introduce experiment setting here, leaving the detailed description of experiment setup, hyper-parameter configurations, model structure, *etc*, to Appendix E.

**Model and Datasets** Most recent papers on robust time series forecasting use LSTM-based structures in their experiments Yoon et al. (2022); Wang et al. (2021a); Wu et al. (2021). Similarly, related works on TSFA tasks also employ LSTM models. Consequently, we primarily use LSTM in our main paper's experiments. However, since our framework is model-agnostic, it can be adapted to other structures like Transformers and TCN, as demonstrated in Appendix F.12. To ensure fairness in comparisons, all methods utilize the same model structure.

We evaluate our methods on the Electricity[1] and Traffic[2] datasets, splitting each dataset into training and test sets with a 7:3 ratio and introducing anomalies into the training set. Notably, we do not create a validation set from the training set for TSFA tasks due to two reasons: 1) Training sets in TSFA are noisy, making validation set selection unreliable for model tuning (as demonstrated in the Appendix F.4); 2) The common practice in LNL suggests that DNN models tend to fit clean samples first and then noisy ones, leading to an initial increase and then decrease in performance, making it insufficient to evaluate TSFA methods solely based on the validation set. Hence, we follow the evaluation protocol from a prominent benchmark method in LNL Li et al. (2020). This protocol records DNN performance on the clean test set at the end of each training epoch, and we report both the **best** and **last** epoch test set performances for each method after training concludes. The datasets are normalized to 0 mean and 1 std before adding anomalies and training.

In the Appendix, we extend our evaluation to more datasets, including real-world time series with intrinsic anomalies (Appendix F.11), and also address the scenario where the test set contains anomalies (Appendix F.1)."

**Comparing methods and evaluation criterion**: We compare our method to the following approaches: **Vanilla training**, which directly trains the model on time series using MSE or MAE loss functions, **Offline imputation (Connor et al., 1994)**, **Online imputation (Bohlke-Schneider et al., 2020)**, and **Loss-based sample selection (Li et al., 2022)**. We use MSE and MAE (see Appendix E for details) on the test set to evaluate each method. It's worth noting that some methods, such as (de Bézenac et al., 2020) and ARIMA (Ho & Xie, 1998), while not designed for TSFA tasks, have shown good performance in scenarios with skewed time series. We provide additional comparisons in the Appendix F.9.

**Training setup:** Each method is trained for 30 epochs using the ADAM optimizer. The learning rate is set to 0.01 for the first 10 epochs and 0.001 for the subsequent 20 epochs. For RobustTSF, we maintain fixed hyperparameters: $\lambda = 0.3$ (Equation (4)), $\tau = 0.3$ (Equation (6)), and $K^{'} = K - 1$ (Dirac weighting) throughout all experiments presented in the paper. It's important to note that our paper primarily focuses on experiments with varying anomaly rates while keeping the anomaly scale constant, in line with conventions in the literature of LNL and related TSFA works. However, we also conduct experiments exploring different anomaly scales, as detailed in Appendix F.7.

---

[1]https://archive.ics.uci.edu/ml/datasets/ElectricityLoadDiagrams 20112014
[2]http://pems.dot.ca.gov

Table 2: Comparison of different methods on Electricity and Traffic dataset for single-step forecasting. We report the best and last epoch test performance for all methods, represented as MAE/MSE. We also report $\Delta = \overline{|\text{best} - \text{last}|}$, which is the average of $|\text{best} - \text{last}|$ for all anomaly settings of each method, to reflect the stableness of each method. The best results are highlighted in bold font. We also establish the statistical consistency of our results, as demonstrated in Appendix F.[17]

| Dataset | Method | | Clean $\eta = 0$ | Constant Anomaly $\eta = 0.1$ | $\eta = 0.3$ | Missing Anomaly $\eta = 0.1$ | $\eta = 0.3$ | Gaussian Anomaly $\eta = 0.1$ | $\eta = 0.3$ | $\Delta$ |
|---|---|---|---|---|---|---|---|---|---|---|
| Electricity | Vanilla (MSE) | best | 0.187/0.071 | 0.199/0.078 | 0.239/0.099 | 0.228/0.097 | 0.305/0.157 | 0.192/0.076 | 0.227/0.096 | 0.019 |
| | | last | 0.205/0.082 | 0.219/0.091 | 0.258/0.117 | 0.229/0.098 | 0.354/0.197 | 0.213/0.087 | 0.255/0.124 | |
| | Vanilla (MAE) | best | 0.182/0.070 | 0.191/0.077 | 0.206/0.086 | 0.205/0.086 | 0.225/0.092 | 0.199/0.080 | 0.210/0.086 | 0.015 |
| | | last | 0.198/0.081 | 0.223/0.096 | 0.245/0.109 | 0.215/0.090 | 0.245/0.107 | 0.203/0.082 | 0.219/0.096 | |
| | Offline | best | 0.184/0.071 | 0.187/0.071 | 0.214/0.087 | 0.190/0.074 | 0.212/0.084 | 0.184/0.070 | 0.193/0.075 | 0.009 |
| | | last | 0.187/0.072 | 0.191/0.073 | 0.232/0.098 | 0.203/0.082 | 0.237/0.100 | 0.186/0.071 | 0.207/0.083 | |
| | Online | best | 0.183/0.070 | 0.199/0.077 | 0.225/0.091 | 0.195/0.075 | 0.227/0.093 | 0.194/0.074 | 0.209/0.084 | 0.01 |
| | | last | 0.201/0.077 | 0.212/0.084 | 0.250/0.104 | 0.215/0.084 | 0.234/0.097 | 0.204/0.078 | 0.215/0.087 | |
| | Loss sel | best | 0.180/0.068 | 0.204/0.081 | 0.209/0.081 | 0.198/0.078 | 0.214/0.089 | 0.196/0.077 | 0.204/0.083 | **0.004** |
| | | last | 0.182/0.069 | 0.208/0.085 | 0.217/0.089 | 0.205/0.083 | 0.217/0.090 | 0.205/0.083 | 0.210/0.086 | |
| | RobustTSF | best | **0.177/0.065** | **0.177/0.066** | **0.183/0.070** | **0.181/0.068** | **0.194/0.074** | **0.176/0.067** | **0.177/0.067** | **0.004** |
| | | last | **0.177/0.066** | **0.182/0.068** | **0.190/0.071** | **0.187/0.071** | **0.204/0.079** | **0.182/0.069** | **0.177/0.068** | |
| Traffic | Vanilla (MSE) | best | 0.196/0.112 | 0.210/0.119 | 0.233/0.129 | 0.268/0.171 | 0.446/0.353 | 0.219/0.132 | 0.292/0.205 | 0.007 |
| | | last | 0.199/0.116 | 0.220/0.126 | 0.235/0.130 | 0.275/0.173 | 0.451/0.357 | 0.233/0.144 | 0.304/0.226 | |
| | Vanilla (MAE) | best | 0.194/0.105 | 0.209/0.123 | 0.231/0.136 | 0.266/0.170 | 0.295/0.208 | 0.216/0.140 | 0.233/0.158 | 0.008 |
| | | last | 0.202/0.114 | 0.210/0.123 | 0.245/0.151 | 0.272/0.186 | 0.307/0.220 | 0.224/0.148 | 0.234/0.159 | |
| | Offline | best | 0.189/0.104 | 0.208/0.123 | 0.222/0.133 | 0.221/0.135 | 0.280/0.186 | 0.205/0.120 | 0.218/0.125 | 0.005 |
| | | last | 0.193/0.110 | 0.220/0.133 | 0.230/0.142 | 0.221/0.135 | 0.281/0.187 | 0.209/0.123 | 0.223/0.131 | |
| | Online | best | 0.206/0.118 | 0.216/0.113 | 0.239/0.126 | 0.220/0.116 | **0.233/0.127** | 0.224/0.127 | 0.244/0.141 | 0.016 |
| | | last | 0.208/0.120 | 0.229/0.136 | 0.248/0.133 | 0.230/0.126 | 0.285/0.177 | 0.234/0.142 | 0.256/0.160 | |
| | Loss sel | best | 0.196/0.115 | 0.226/0.144 | 0.257/0.160 | 0.241/0.153 | 0.325/0.235 | 0.211/0.133 | 0.260/0.179 | 0.004 |
| | | last | 0.199/0.120 | 0.232/0.151 | 0.260/0.170 | 0.241/0.153 | 0.329/0.238 | 0.220/0.138 | 0.260/0.179 | |
| | RobustTSF | best | **0.185/0.100** | **0.198/0.113** | **0.203/0.111** | **0.200/0.113** | 0.243/0.140 | **0.185/0.101** | **0.202/0.116** | **0.003** |
| | | last | **0.185/0.100** | **0.200/0.112** | **0.209/0.116** | **0.203/0.114** | 0.250/0.152 | **0.185/0.101** | **0.205/0.119** | |

Table 3: Comparison of different methods on Electricity dataset for multi-step forecasting. We report the best epoch performance for all the methods by MAE/MSE. The anomaly ratio for each anomaly type is 0.2. The prediction length is 4 and 8. The best results are highlighted in bold font.

| Method | Clean $O = 4$ | Constant ($\eta = 0.2$) $O = 4$ | $O = 8$ | Missing ($\eta = 0.2$) $O = 4$ | $O = 8$ | Gaussian ($\eta = 0.2$) $O = 4$ | $O = 8$ |
|---|---|---|---|---|---|---|---|
| Vanilla (MSE) | 0.302/0.175 | 0.321/0.191 | 0.384/0.262 | 0.367/0.229 | 0.423/0.298 | 0.324/0.191 | 0.369/0.241 |
| Vanilla (MAE) | 0.292/0.165 | 0.302/0.174 | 0.382/0.264 | 0.327/0.192 | 0.394/0.283 | 0.310/0.181 | 0.371/0.247 |
| Offline | 0.302/0.179 | 0.314/0.184 | 0.399/0.286 | 0.334/0.216 | 0.432/0.358 | 0.312/0.188 | 0.407/0.306 |
| Online | 0.308/0.188 | 0.306/0.176 | 0.413/0.308 | 0.337/0.213 | 0.477/0.412 | 0.346/0.216 | 0.471/0.391 |
| Loss sel | 0.304/0.184 | 0.319/0.197 | 0.383/0.284 | 0.342/0.217 | 0.412/0.318 | 0.319/0.195 | 0.381/0.275 |
| RobustTSF | **0.280/0.158** | **0.292/0.163** | **0.371/0.252** | **0.314/0.185** | **0.371/0.249** | **0.284/0.156** | **0.353/0.234** |

## 7.1 SINGLE-STEP TIME SERIES FORECASTING

We first perform experiments on single-step forecasting following exact problem formulation in Section 3. The length of input sequence is 16 for Electricity and Traffic. The overall results are reported in Table 2 from which we can observe some interesting phenomenons:

1) Among all anomaly types, missing anomalies have the most significant negative impact on model performance 2) Offline imputation, online imputation, and loss selection approaches do not consistently improve performance across all anomaly types within each dataset. For instance, the online imputation method improves performance for the missing anomaly type in the Traffic dataset but decreases it for constant and Gaussian anomalies. This phenomenon also occurs with loss-based sample selection. The difference arises because, unlike LNL, TSFA involves noise (anomalies) in both $X$ and $Y$, making the small loss criterion in LNL less suitable for TSFA tasks. 3) Loss selection and RobustTSF are the most stable methods (with small $\Delta$ values). Both approaches select reliable training samples, but RobustTSF outperforms the loss selection approach. 4) RobustTSF consistently improves performance compared to vanilla training across all anomaly types and datasets. This suggests the effectiveness of our selection module. Notably, even without manually adding anomalies (clean), RobustTSF still exhibits clear improvements. This is because sensor-recorded data inevitably contain anomalies. Thus, our method serves as a general approach for time series forecasting tasks.

## 7.2 GENERALIZATION TO MULTI-STEP FORECASTING

Our approach can be easily generalized to multi-forecasting tasks, *i.e.,*. predicting multi-step values given input time series. For this part of experiment, we evaluate RobustTSF on Electricity dataset. The input length is fixed to be 96 and the prediction length $O \in \{4, 8\}$. The best epoch performance for each method is reported in Table 3. We can observe that RobustTSF still has consistent improvement compared to the other baseline methods, including the setting of clean time series.

### 7.3 GENERALIZATION TO OTHER ANOMALIES

Even though our theories in Section 3 are based on three common point-level anomalies, i.e., Constant, Missing and Gaussian Anomalies, RobustTSF works well when anomalies follow other heavy-tailed distributions like student-t distribution and Generalized Pareto distribution. The corresponding evaluations are left in Appendix F.10. Furthermore, RobustTSF can also be adopted in case of subsequence anomaly. Considering a simple subsequence anomaly modeled by Markov process:

$$\text{If } \widetilde{z}_{t-1} \text{ is clean}, \widetilde{z}_t = \begin{cases} z_t, & \text{with probability } 1 - \eta \\ z_t^{\mathrm{A}}, & \text{with probability } \eta \end{cases} \tag{7}$$

$$\text{If } \widetilde{z}_{t-1} \text{ is anomaly}, \widetilde{z}_t = \begin{cases} z_t, & \text{with probability } 1 - \phi \\ \widetilde{z}_{t-1} + \epsilon, \ \epsilon \sim \mathcal{N}(0, 0.1), & \text{with probability } \phi \end{cases} \tag{8}$$

Equation (7) behaves like a point-level anomaly if the previous time step is clean. However, when the previous time step is an anomaly, the property of subsequence anomaly suggests that the current time step is likely to also be an anomaly, with a value close to the previous anomaly (Equation (8)). We find that RobustTSF performs well in this scenario. This is because our filtering scheme (Equation (6)) primarily selects samples with clean time steps near the label, allowing anomalies to be modeled by Equation (7) after sample selection. Empirical experiments in Table 4 confirm this observation, as RobustTSF outperforms other methods in the subsequence anomaly setting. It's worth noting that for more complex anomaly types that may not satisfy $\mathbb{P}(\widetilde{Y}|Y) \neq \mathbb{P}(\widetilde{Y}|Y, X)$, MAE may not be theoretically robust. Analyzing and developing loss functions for these anomaly types presents an interesting avenue for future research.

Table 4: Comparison of different methods on Electricity dataset for single-step forecasting with subsequence anomaly. We report the best epoch performance for all methods by MAE/MSE. We fix $\eta = 0.3$ in Equation (7) and vary $\phi$ in Equation (8). The best results are highlighted in bold font.

| Method | Gaussian ($\eta = 0.3$) | | | | |
|---|---|---|---|---|---|
| | $\phi = 0.1$ | $\phi = 0.3$ | $\phi = 0.5$ | $\phi = 0.7$ | $\phi = 0.9$ |
| Vanilla (MSE) | 0.206/0.083 | 0.214/0.089 | 0.253/0.111 | 0.337/0.209 | 0.479/0.388 |
| Vanilla (MAE) | 0.201/0.080 | 0.193/0.076 | 0.194/0.076 | 0.202/0.081 | 0.206/0.084 |
| Offline | 0.194/0.075 | 0.194/0.075 | 0.191/0.074 | 0.198/0.078 | 0.203/0.083 |
| Online | 0.196/0.075 | 0.197/0.077 | 0.197/0.075 | 0.197/0.077 | 0.200/0.079 |
| Loss sel | 0.198/0.075 | 0.199/0.079 | 0.199/0.079 | 0.202/0.082 | 0.203/0.081 |
| RobustTSF | **0.179/0.067** | **0.180/0.068** | **0.185/0.072** | **0.187/0.073** | **0.198/0.079** |

### 7.4 HYPER-PARAMETER TUNING AND DISCUSSIONS

To further study the effect of each component of RobustTSF, we conduct Hyper-parameter Tuning towards the loss, weighting strategy and anomaly detection method shown in Table 5, showing the advantage of each component of our design. More discussions (including test set with anomalies, forecasting visualization, sensitivity of detection-imputation-retraining pipeline, parameter sensitivity, efficiency analyses, RobustTSF on more datasets, RobustTSF on Transformer and TCN models, comparing RobustTSF with more methods, more discussion on RobustTSF such as significance and future work, *etc*) are summarized in Appendix F.

Table 5: Hyper-parameterTuning of RobustTSF on Electricity dataset. We report the best epoch performance (MAE) for all methods. The best results are highlighted in bold font.

| Method | Const | | Missing | | Gaussian | |
|---|---|---|---|---|---|---|
| | $\eta = 0.1$ | $\eta = 0.3$ | $\eta = 0.1$ | $\eta = 0.3$ | $\eta = 0.1$ | $\eta = 0.3$ |
| RobustTSF (MAE, Dirac weighting, trend filter detection) | **0.177** | **0.183** | **0.181** | **0.194** | **0.176** | **0.177** |
| RobustTSF (MSE, Dirac weighting, trend filter detection) | 0.180 | 0.193 | 0.182 | 0.249 | 0.177 | 0.181 |
| RobustTSF (MAE, exponential weighting, trend filter detection) | 0.178 | 0.183 | 0.182 | 0.199 | 0.179 | 0.180 |
| RobustTSF (MAE, Dirac weighting, prediction based detection) | 0.191 | 0.217 | 0.195 | 0.206 | 0.180 | 0.184 |

## 8 CONCLUSIONS

This paper aims to deal with TSFA (time series forecasting with anomaly) tasks. We first define common types of anomalies and analyze the loss robustness and sample robustness on these anomaly types. Then based on our analyses, we develop an efficient algorithm to improve model performance on TSFA tasks and achieve SOTA performance. Furthermore, this is the first study to form a bridge between LNL task and TSFA task and may potentially inspire future research on this topic.

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

APPENDIX ARRANGEMENT

The appendix is arranged as follows:

## A  PROOF FOR THEOREMS

### A.1  PROOF FOR THEOREM 1

Let $R_\ell(f) = \mathbb{E}_\mathcal{D}\ell(f(X), Y) = \mathbb{E}_{\boldsymbol{x},y_{\boldsymbol{x}}}\ell(f(\boldsymbol{x}), y_{\boldsymbol{x}})$ and $R_\ell^\eta(f) = \mathbb{E}_{\widetilde{\mathcal{D}}}\ell(f(X), \widetilde{Y}) = \mathbb{E}_{\boldsymbol{x},\widetilde{y}_{\boldsymbol{x}}}\ell(f(\boldsymbol{x}), \widetilde{y}_{\boldsymbol{x}})$, respectively. Then

$$
\begin{aligned}
&R_\ell^\eta(f) \\
=& \mathbb{E}_{\boldsymbol{x},\widetilde{y}_{\boldsymbol{x}}}\ell(f(\boldsymbol{x}), \widetilde{y}_{\boldsymbol{x}}) \\
=& \mathbb{E}_{\boldsymbol{x}}\mathbb{E}_{y_{\boldsymbol{x}}|\boldsymbol{x}}\mathbb{E}_{\widetilde{y}_{\boldsymbol{x}}|\boldsymbol{x},y_{\boldsymbol{x}}}\ell(f(\boldsymbol{x}), \widetilde{y}_{\boldsymbol{x}}) \\
=& \mathbb{E}_{\boldsymbol{x}}\mathbb{E}_{y_{\boldsymbol{x}}|\boldsymbol{x}}[(1-\eta)\cdot\ell(f(\boldsymbol{x}), y_{\boldsymbol{x}}) + \eta\cdot\ell(f(\boldsymbol{x}), y_{\boldsymbol{x}}^{\mathrm{A}})] \\
=& \mathbb{E}_{\boldsymbol{x}}\mathbb{E}_{y_{\boldsymbol{x}}|\boldsymbol{x}}[(1-\eta)\cdot\ell(f(\boldsymbol{x}), y_{\boldsymbol{x}}) + \eta\cdot(C_{\boldsymbol{x}} - \ell(f(\boldsymbol{x}), y_{\boldsymbol{x}}))] \\
=& (1 - 2\cdot\eta)\cdot R_\ell(f) + \eta\cdot\mathbb{E}_{\boldsymbol{x},y_{\boldsymbol{x}}}C_{\boldsymbol{x}}
\end{aligned}
$$

where $\gamma_1 = 1 - 2\cdot\eta > 0$ and $\gamma_2 = \eta\cdot\mathbb{E}_{x,y_{\boldsymbol{x}}}C_{\boldsymbol{x}}$, which are constants respect to $f$.

### A.2  PROOF FOR PROPOSITION 1

Let $q = f(\boldsymbol{x})$, then we are validating the value of $\ell(q, y_{\boldsymbol{x}}) + \ell(q, y_{\boldsymbol{x}}^{\mathrm{A}})$. For MAE loss, the value is represented as $|q - y_{\boldsymbol{x}}| + |q - y_{\boldsymbol{x}}^{\mathrm{A}}|$. From the property of absolute function, $|q - y_{\boldsymbol{x}}| + |q - y_{\boldsymbol{x}}^{\mathrm{A}}|$ achieves minimum when $q$ lies in the range between $y_{\boldsymbol{x}}$ and $y_{\boldsymbol{x}}^{\mathrm{A}}$. Thus when $\min\{y_{\boldsymbol{x}}, y_{\boldsymbol{x}}^{\mathrm{A}}\} < q < \max\{y_{\boldsymbol{x}}, y_{\boldsymbol{x}}^{\mathrm{A}}\}$, we have $|q - y_{\boldsymbol{x}}| + |q - y_{\boldsymbol{x}}^{\mathrm{A}}| = C_{\boldsymbol{x}} = |y_{\boldsymbol{x}} - y_{\boldsymbol{x}}^{\mathrm{A}}|$.

This conclusion can not be fit for MSE loss, since $(q - y_{\boldsymbol{x}})^2 + (q - y_{\boldsymbol{x}}^{\mathrm{A}})^2$ is not constant when $\min\{y_{\boldsymbol{x}}, y_{\boldsymbol{x}}^{\mathrm{A}}\} < q < \max\{y_{\boldsymbol{x}}, y_{\boldsymbol{x}}^{\mathrm{A}}\}$.

### A.3 PROOF FOR PROPOSITION 2

Following the notation in Section A.2, it is easy to verify that $\ell_{\text{norm}}(q, y_{\boldsymbol{x}}) + \ell_{\text{norm}}(q, y_{\boldsymbol{x}}^{\text{A}}) = 1$, which is constant respect to $f$.

### A.4 PROOF FOR THEOREM 2

Following the notation in the proof for Theorem 1, we aim to show that $\mathbb{E}_{\boldsymbol{x}}\mathbb{E}_{y_{\boldsymbol{x}}|\boldsymbol{x}}[(1-\eta)\cdot\ell(f(\boldsymbol{x}), y_{\boldsymbol{x}}) + \eta \cdot \ell(f(\boldsymbol{x}), y_{\boldsymbol{x}}^{\text{A}})]$, $y_{\boldsymbol{x}}^{\text{A}} \sim \mathcal{N}(y_{\boldsymbol{x}}, \sigma^2)$ achieves minimum when $f(\boldsymbol{x}) = y_{\boldsymbol{x}}$.

• For MSE loss, let $q = f(\boldsymbol{x})$, then we are to show that $(1 - \eta) \cdot (q - y_{\boldsymbol{x}})^2 + \eta \cdot (q - y_{\boldsymbol{x}}^{\text{A}})^2$ achieves minimum when $q = y_{\boldsymbol{x}}$. Since $(1 - \eta) \cdot (q - y_{\boldsymbol{x}})^2$ achieves minimum when $q = y_{\boldsymbol{x}}$, we only need to prove $\int_{-\infty}^{+\infty} g(y_{\boldsymbol{x}}^{\text{A}}) \cdot (y_{\boldsymbol{x}}^{\text{A}} - q)^2 dy_{\boldsymbol{x}}^{\text{A}}$ achieves minimum when $q = y_{\boldsymbol{x}}$ where $g(y_{\boldsymbol{x}}^{\text{A}}) = \frac{1}{\sqrt{2\pi}\sigma}\exp(-\frac{(y_{\boldsymbol{x}}^{\text{A}}-y_{\boldsymbol{x}})^2}{2\sigma^2})$ is the pdf of the distribution of $y_{\boldsymbol{x}}^{\text{A}}$.

Let $s(q) = \mathbb{E}_{y_{\boldsymbol{x}}^{\text{A}}}(y_{\boldsymbol{x}}^{\text{A}} - q)^2 = \int_{-\infty}^{+\infty} g(y_{\boldsymbol{x}}^{\text{A}}) \cdot (y_{\boldsymbol{x}}^{\text{A}} - q)^2 dy_{\boldsymbol{x}}^{\text{A}}$, then

$$\frac{\partial s(q)}{\partial q} = 0$$
$$\Longrightarrow 2 \cdot q - 2 \cdot \mathbb{E}_{y_{\boldsymbol{x}}^{\text{A}}} g(y_{\boldsymbol{x}}^{\text{A}}) = 0$$
$$\Longrightarrow q = \mathbb{E}_{y_{\boldsymbol{x}}^{\text{A}}} g(y_{\boldsymbol{x}}^{\text{A}}) = y_{\boldsymbol{x}}$$

Thus $\int_{-\infty}^{+\infty} g(y_{\boldsymbol{x}}^{\text{A}}) \cdot (y_{\boldsymbol{x}}^{\text{A}} - q)^2 dy_{\boldsymbol{x}}^{\text{A}}$ achieves minimum when $q = y_{\boldsymbol{x}}$.

• MAE loss can also be proved with a similar procedure as follows.

Let $s(q) = \mathbb{E}_{y_{\boldsymbol{x}}^{\text{A}}}|y_{\boldsymbol{x}}^{\text{A}} - q| = \int_{-\infty}^{+\infty} g(y_{\boldsymbol{x}}^{\text{A}}) \cdot |y_{\boldsymbol{x}}^{\text{A}} - q| dy_{\boldsymbol{x}}^{\text{A}}$, then

$$\frac{\partial s(q)}{\partial q} = 0$$
$$\Longrightarrow \mathbb{E}_{y_{\boldsymbol{x}}^{\text{A}}}(\frac{-1 \cdot (y_{\boldsymbol{x}}^{\text{A}} - q)}{|y_{\boldsymbol{x}}^{\text{A}} - q|}) = 0$$
$$\Longrightarrow \mathbb{E}_{y_{\boldsymbol{x}}^{\text{A}}}(\mathbb{1}\{y_{\boldsymbol{x}}^{\text{A}} < q\} - \mathbb{1}\{y_{\boldsymbol{x}}^{\text{A}} > q\}) = 0$$
$$\Longrightarrow \mathbb{P}(y_{\boldsymbol{x}}^{\text{A}} < q) = \mathbb{P}(y_{\boldsymbol{x}}^{\text{A}} > q)$$
$$\Longrightarrow q = y_{\boldsymbol{x}}$$

Thus $\int_{-\infty}^{+\infty} g(y_{\boldsymbol{x}}^{\text{A}}) \cdot |y_{\boldsymbol{x}}^{\text{A}} - q| dy_{\boldsymbol{x}}^{\text{A}}$ achieves minimum when $q = y_{\boldsymbol{x}}$.

## B VISUALIZATION FOR COMMON ANOMALY TYPES

As discussed in Section 3, three types of anomalies are considered in this paper. Figure 2 shows visualization examples for these anomaly types.

## C EXPERIMENTS ON LOSS ROBUSTNESS

In Section 4, we provided our Theorems on loss robustness when there are anomalies in $Y$. Here, we examine our Theorems in two real-world time series datasets: Electricity and Traffic, which are used by many time series forecasting papers (Yoon et al., 2022; Wang et al., 2021a; Wu et al., 2021). Since many works (Connor et al., 1994; Bohlke-Schneider et al., 2020; Li et al., 2022) dealing with TSFA problems deploy RNN-based structure to conduct experiments, we choose to use LSTM to validate our Theorems. Experiments are shown in Table 6. It can be observed that for Constant and Missing type anomalies, MAE is very robust whose performance does not vary too much when anomalies exist while MSE degrades the performance with the increase of the anomaly rate. For Gaussian-type anomalies, MAE and MSE can both be robust. These results support our theoretical analyses for loss robustness when anomalies only exist in the label.

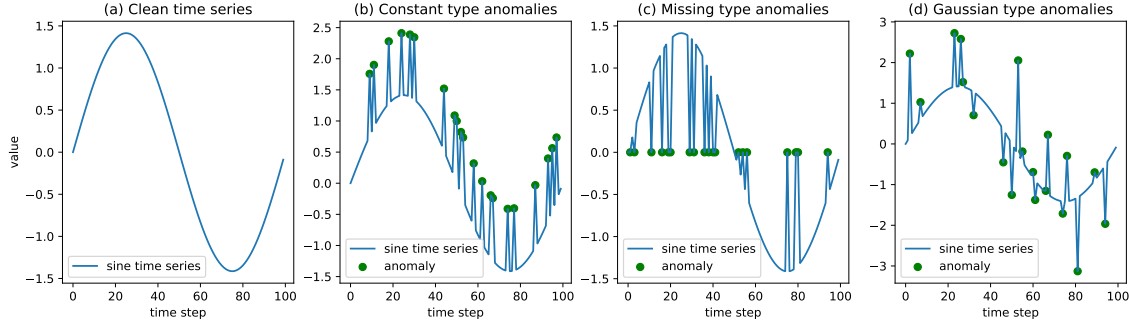

Figure 2: Visualization of time series with different types of anomalies. (a): Clean time series which is the sine function of time steps. We normalize the time series to 0 mean and 1 std. (b) (c) (d): Time series (sine) with Constant, Missing, and Gaussian type anomalies. The noise rate for these types of anomalies is 0.2. The noise scale is 1.0 for Constant and Gaussian anomaly and 0 for Missing anomaly.

Table 6: Comparison of loss functions on different anomaly types with anomaly rate 0.1 and 0.2. For each setting, we report the best performance on test set, presented as MAE/MSE. The model structure is LSTM. Detailed experimental setting is left in Appendix E.

| Dataset | Loss | *Constant Anomaly* $\eta = 0.1$ | $\eta = 0.2$ | *Missing Anomaly* $\eta = 0.1$ | $\eta = 0.2$ | *Gaussian Anomaly* $\eta = 0.1$ | $\eta = 0.2$ |
|---------|------|------|------|------|------|------|------|
| Electricity | MAE | 0.185/0.070 | 0.187/0.071 | 0.189/0.071 | 0.195/0.078 | 0.184/0.070 | 0.186/0.073 |
| | MSE | 0.190/0.071 | 0.205/0.081 | 0.209/0.082 | 0.217/0.089 | 0.187/0.071 | 0.188/0.074 |
| Traffic | MAE | 0.194/0.108 | 0.199/0.112 | 0.206/0.119 | 0.210/0.123 | 0.198/0.112 | 0.206/0.117 |
| | MSE | 0.208/0.119 | 0.222/0.124 | 0.249/0.148 | 0.302/0.191 | 0.204/0.121 | 0.214/0.131 |

Table 7: Evaluating how different positions of anomalies in the input time series affect prediction performance. For each dataset, we simulate different types of anomalies with anomaly rates 0.1 and 0.2. The loss is MAE and the network structure is LSTM. we report best epoch MAE for each setting

| Anomaly Type | ETT-h1 with Anomaly Position front | middle | back | Exchange with Anomaly Position front | middle | back |
|--------------|-------|--------|------|-------|--------|------|
| Clean | 0.051 | 0.051 | 0.051 | 0.047 | 0.047 | 0.047 |
| Const. ($\eta = 0.1$) | 0.051 | 0.051 | 0.052 | 0.047 | 0.049 | 0.052 |
| Const. ($\eta = 0.2$) | 0.052 | 0.051 | 0.055 | 0.049 | 0.050 | 0.060 |
| Missing. ($\eta = 0.1$) | 0.052 | 0.052 | 0.054 | 0.049 | 0.054 | 0.173 |
| Missing. ($\eta = 0.2$) | 0.052 | 0.052 | 0.058 | 0.050 | 0.050 | 0.214 |
| Gaussian. ($\eta = 0.1$) | 0.051 | 0.051 | 0.053 | 0.048 | 0.049 | 0.057 |
| Gaussian. ($\eta = 0.2$) | 0.052 | 0.052 | 0.058 | 0.050 | 0.051 | 0.062 |

## D    MORE EXPERIMENTS ON SAMPLE ROBUSTNESS

We show more experiments regarding to Sample Robustness to support our hypothesis in Section 5. The datasets used are ETT-h1 (Zhou et al., 2021) and Exchange (Lai et al., 2018). The results in Table 7 demonstrates minimal performance drop for distant anomalies, consistent with Section 5.

## E    DETAILED EXPERIMENTAL SETTING

**Model structure:** We use the same LSTM model structure to implement all the methods for fair comparison. Note that all the methods in Table 2 are model agnostic, except for online imputation which needs a specific auto-regressive RNN-based model. The LSTM we use in the paper (Table 6, Table 1, Table 2, Table 3, Table 4, Table 12) consists of two recurrent layers with the hidden size 10 and a fully connected layer.

The Transformer model we use in the paper (Table 24) consists of a Positional Encoding layer with feature size 250, a vanilla transformer encoding layer with 10 heads, and a fully connected layer.

**Dataset preprocessing:** We first normalize the clean training set to 0 mean and 1 std and add anomalies. When adding constant type anomaly, the noise scale is $\epsilon = 0.5 \cdot std = 0.5$; when adding

missing type anomaly, the noise scale is $\epsilon = 0$; when adding Gaussian type anomaly, the noise scale is $\epsilon = 2 \cdot \text{std} = 2$; The clean test set is also scaled using the scaling parameter of training set when evaluating each method, i.e., $z_{\text{test}} = \frac{z_{\text{test}} - \mu}{\delta}$, where $\mu$ and $\delta$ are the mean and std of the training set, respectively.

**Evaluation:** We use MAE and MSE on the test set to evaluate each method:

$$\text{MAE} = \frac{1}{n} \sum_{i=1}^{n} |y - \hat{y}| \ , \ \text{MSE} = \frac{1}{n} \sum_{i=1}^{n} (y - \hat{y})^2$$

**Shared hyper-parameter for all the methods** We train each method for 30 epochs and report the best epoch test performance and last epoch test performance. The learning rate is 0.1 for the first 10 epochs and 0.01 for the last 20 epochs. The optimizer is Adam and the batch size is 128.

**Specific hyper-parameter for offline and online imputation** Since detection-imputation-retraining pipeline requires a threshold to determine whether each time step is an anomaly step and this pipeline can be sensitive to anomaly types (Section F.5). When training detection-imputation-retraining pipeline on different anomaly types, we choose the threshold from {0.5,0.6,0.7,0.8} to get the best result.

**Specific hyper-parameter for loss-based sample selection:** For each anomaly type, we pretrain the model for 3 epochs and record the loss of all the samples after each training epoch ends. Then we calculate the mean and std of the loss for all the samples and select the samples with small loss and small loss std. Note (Li et al., 2022) assumes the anomaly ratio is known. Thus we select (1 - anomaly ratio)· Num. of Samples.

**Specific hyper-parameter for RobustTSF:** We fix $\lambda = 0.3$ (Equation (4)), $\tau = 0.3$ (Equation (6)), $K' = K - 1$ (Dirac weighting scheme, where $K$ is the length of the input time series) for all the experiments in the paper.

# F  MORE EXPERIMENTS AND DISCUSSIONS ON ROBUSTTSF

## F.1  APPLYING ROBUSTTSF WHEN TEST SET ALSO HAS ANOMALIES

In the main paper, we assume the test set is clean without anomalies by following the setting of learning with noisy labels and related works on TSFA tasks (Connor et al., 1994). However, the test set may also have anomalies in the real-world scenario. Here we show how to apply RobustTSF when the test set is noisy.

Assume anomalies exist in $X$ in the test set. Since $Y$ is used to justify the performance of each method, noisy $Y$ cannot determine if the method is learning towards the right target. When performing testing, we first use a modified trend filter (Equation (5) in the main paper) to calculate the trend of each input $\boldsymbol{x}$. If the amount of anomaly in the last time step is smaller than $\tau$, then we do not modify $\boldsymbol{x}$. Otherwise, we use the last value of the trend to replace the value of the last time step in $\boldsymbol{x}$. Then we use the trained model to evaluate the test set. We report the performance (best epoch MAE) on the Electricity dataset shown in Table 8. It can be observed that RobustTSF outperforms other methods on test sets with anomalies, providing valuable insights for practical applications.

Table 8: Comparing RobustTSF with other methods on noisy test set anomalies

| Method | Constant | | Missing | | Gaussian | |
|---|---|---|---|---|---|---|
| | $\eta = 0.01$ | $\eta = 0.05$ | $\eta = 0.01$ | $\eta = 0.05$ | $\eta = 0.01$ | $\eta = 0.05$ |
| MAE | 0.192 | 0.253 | 0.201 | 0.313 | 0.200 | 0.258 |
| MSE | 0.211 | 0.281 | 0.237 | 0.387 | 0.212 | 0.275 |
| Offline | 0.190 | 0.232 | 0.198 | 0.324 | 0.191 | 0.255 |
| Online | 0.197 | 0.270 | 0.202 | 0.362 | 0.212 | 0.250 |
| Loss sel | 0.207 | 0.235 | 0.208 | 0.311 | 0.199 | 0.236 |
| RobustTSF | **0.174** | **0.219** | **0.183** | **0.302** | **0.178** | **0.190** |

## F.2 Visualizing Forecasting Results of Each Method

In this part, we visualize the forecasting results of each method to see if RobustTSF can learn intrinsic patterns within the data. Each method is trained on the Electricity dataset with Gaussian noise ($\eta = 0.3$) and tested on the clean test set. The results are shown in Figure 3. It can be observed that only RobustTSF can learn underlying patterns within the time series.

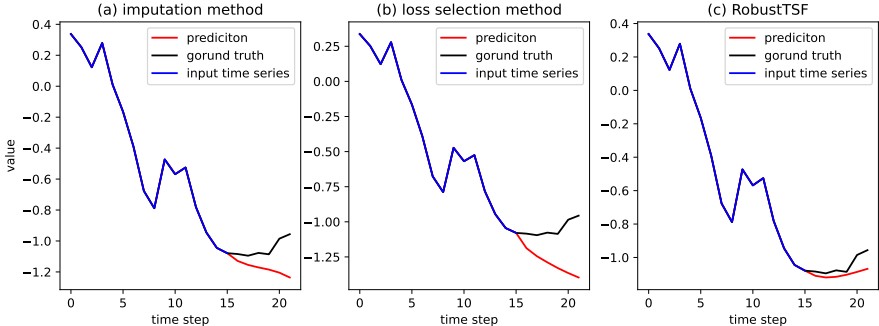

Figure 3: (a) The forecasting result of offline imputation. (b) The forecasting result of loss-based sample selction. (c) The forecasting result of RobustTSF. The length of input time series is 16 and the forecasting horizon is 4.

## F.3 More Subsequence Anomaly Experiment and Multi-step Forecasting Experiment

In the main paper, we show the result of Gaussian-based subsequence anomaly. We provide experiments of constant and missing type-based subsequence anomalies in this section. Results are shown in Table 9. It can be observed that RobustSTF is still superior to the other methods.

Table 9: Comparison of different methods on Electricity dataset for single-step forecasting with subsequence anomaly. We report the best epoch performance for all methods, represented as MAE/MSE. We fix $\eta = 0.3$ in Equation (7) and vary $\phi$ in Equation (8). The best results are highlighted in bold font.

| Method | Constant ($\eta = 0.3$) | | | Missing ($\eta = 0.3$) | | |
|---|---|---|---|---|---|---|
| | $\phi = 0.5$ | $\phi = 0.7$ | $\phi = 0.9$ | $\phi = 0.5$ | $\phi = 0.7$ | $\phi = 0.9$ |
| Vanilla (MAE) | 0.207/0.082 | 0.210/0.083 | 0.210/0.085 | 0.213/0.090 | 0.212/0.085 | 0.202/0.081 |
| Offline | 0.202/0.079 | 0.208/0.083 | 0.210/0.085 | 0.206/0.084 | 0.209/0.084 | 0.202/0.082 |
| Online | 0.216/0.087 | 0.215/0.084 | 0.208/0.082 | 0.285/0.149 | 0.295/0.156 | 0334/0.220 |
| Loss sel | 0.204/0.081 | 0.208/0.084 | 0.209/0.085 | 0.213/0.088 | 0.210/0.086 | 0.237/0.104 |
| RobustTSF | **0.188/0.072** | **0.194/0.077** | **0.200/0.081** | **0.192/0.074** | **0.192/0.074** | **0.193/0.075** |

For the multi-step forecasting, we use another setting to perform experiments with input length of 12 and prediction length of 12. The experiments on Electricity dataset are presented in Table 10. The results affirm that RobustTSF maintains good performance with input and output length of 12.

## F.4 Model Selection via Noisy Validation Set

In this section, we aim to show that the noisy validation is not reliable for selecting models. We split the Electricity data into separate training, validation, and testing sets with a ratio of 6:2:2 and add the anomalies in the training set and validation set. The best epoch test accuracy and the test accuracy using the model selected by the validation set are reported. The results are shown in Table 11. It can be observed that $\Delta$ is relatively large whose value is even similar to $\overline{|\text{best} - \text{last}|}$ in Table 2. Thus using the noisy validation set to select the model is not reliable.

Table 10: Comparing RobustTSF with other methods for multi-step forecasting with input and output length of 12. Best epoch MAE is reported.

| Method | Constant | | Missing | | Gaussian | |
|---|---|---|---|---|---|---|
| | $\eta = 0.1$ | $\eta = 0.3$ | $\eta = 0.1$ | $\eta = 0.3$ | $\eta = 0.1$ | $\eta = 0.3$ |
| MAE | 0.550 | 0.575 | 0.544 | 0.572 | 0.541 | 0.565 |
| MSE | 0.551 | 0.578 | 0.545 | 0.595 | 0.545 | 0.568 |
| Offline | 0.539 | 0.552 | 0.527 | 0.616 | 0.531 | 0.546 |
| Online | 0.623 | 0.627 | 0.617 | 0.680 | 0.605 | 0.654 |
| Loss sel | 0.560 | 0.564 | 0.535 | 0.623 | 0.549 | 0.563 |
| RobustTSF | **0.512** | **0.543** | **0.525** | **0.547** | **0.516** | **0.523** |

Table 11: Validating the reliability of noisy validation on Electricity dataset. **best** represents the best epoch test accuracy, **val_best** represents the test accuracy using the model selected by the validation set. The performance is represented as MAE/MSE. We also report $\Delta = \overline{|best - val\_best|}$ to reflect the reliability of using the noisy validation set to select the models.

| Method | | Constant Anomaly | | Missing Anomaly | | Gaussian Anomaly | | |
|---|---|---|---|---|---|---|---|---|
| | | $\eta = 0.2$ | $\eta = 0.3$ | $\eta = 0.2$ | $\eta = 0.3$ | $\eta = 0.2$ | $\eta = 0.3$ | $\Delta$ |
| Vanilla | best | 0.217/0.086 | 0.222/0.097 | 0.234/0.094 | 0.240/0.104 | 0.205/0.083 | 0.203/0.078 | 0.014 |
| (MAE) | val_best | 0.238/0.101 | 0.252/0.112 | 0.247/0.107 | 0.251/0.111 | 0.218/0.090 | 0.221/0.092 | |

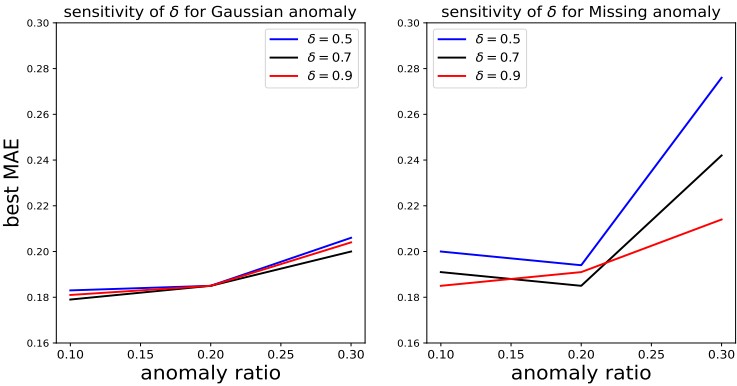

Figure 4: Sensitivity of $\delta$ with Gaussian and Missing anomaly for offline detection-imputation-retraining pipeline.

### F.5 SENSITIVITY OF DETECTION-IMPUTATION-RETRAINING PIPELINE

The most important hyper-parameter for detection-imputation-retraining pipeline (Connor et al., 1994; Bohlke-Schneider et al., 2020; Li et al., 2022) is the threshold for determining whether each time step is an anomaly, *i.e.,* if $\hat{z}_i - \widetilde{z}_i > \delta$, where $\hat{z}_i$ is the model prediction at time step $i$, $\widetilde{z}_i$ is the original observed value at time step $i$ and $\delta$ is the threshold. We conduct an experiment on Electricity dataset shown in Figure 4. It can be observed that for Gaussian anomaly, different values of $\delta$ result in similar performances and $\delta = 0.7$ performs slightly better than others. However, for Missing type anomaly, values of $\delta$ can make performance vary very much and $\delta = 0.9$ performs better than others. This part of experiment shows that the hyper-parameter of detection-imputation-retraining pipeline is sensitive to anomaly type. Recall that for RobustTSF, we fix all the hyper-parameters for all the settings. Thus RobustTSF is more robust to hyper-parameters.

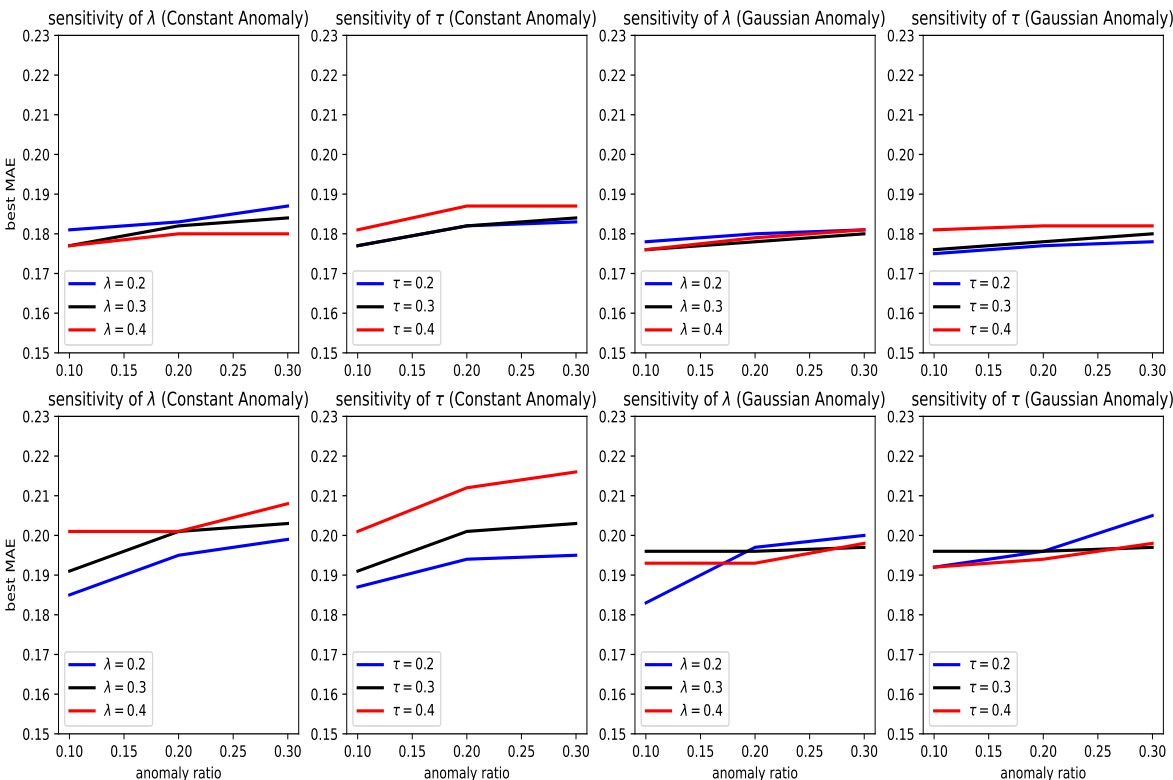

Figure 5: The figure illustrates experiments concerning the hyperparameters $\tau$ and $\lambda$ for RobustTSF on both the Electricity and Traffic datasets, each containing different types of anomalies. The first row of subfigures corresponds to the Electricity dataset, while the second row corresponds to the Traffic dataset. Notably, the results indicate that RobustTSF remains robust across various settings of hyperparameters. We reiterate that we maintain consistent hyperparameters across all settings when comparing with other methods.

## F.6 PARAMETER SENSITIVITY ANALYSES

To further study the influence of the hyper-parameters of RobustTSF, we conduct the parameter sensitivity analyses on the Electricity and Traffic dataset. The results can be observed in Figure 5. It can be seen that RobustTSF is quite robust to these hyper-parameters. Each setting can outperform other baseline methods.

We also perform an experiment to show the influence of different sample selection metric shown in Table 12. It can be observed that random selection makes performance worse while all the weighting schemes, which are all non-decreasing function with respect to the time step, have similar performance on TSFA tasks.

## F.7 EXPERIMENTS WITH RESPECT TO ANOMALY SCALES

There exists three hyper-parameters, namely $\eta$, $\epsilon$ and $\gamma$ characterizing three anomaly types we defined in the paper. In our main paper, we have concentrated our experiments on the $\eta$ hyper-parameter, which aligns with the conventions found in the literature of LNL and related TSFA works. This choice is rooted in the fact that even the original TSFA paper Connor et al. (1994) employed a fixed $\gamma^2$ value throughout their experiments.

For both $\epsilon$ and $\gamma$, we have extended our experimentation efforts, as reflected in Figure 6. Specifically, Figure 6 (a) and (c) delineate the $\epsilon$ experimentation for the Constant anomaly, while Figure 6 (b) and (d) illustrate the $\epsilon$ experiment for the Gaussian anomaly. It's worth noting that in the context of Gaussian anomaly, the value of $\gamma^2$ characterizes the distribution of $\epsilon$. In Figure 6 (b) and (d),

Table 12: Comparison of different sample selection metric for RobustTSF. 'Random' means we randomly select $N * (1 - \text{anomaly ratio})$ samples for training. We report the best epoch performance for all the methods, represented as MAE/MSE.

| Method | Constant $\eta = 0.3$ | Missing $\eta = 0.3$ | Gaussian $\eta = 0.3$ |
|---|---|---|---|
| Random | 0.222/0.089 | 0.211/0.086 | 0.201/0.079 |
| Exponential weighting | 0.183/0.070 | 0.196/0.075 | 0.179/0.067 |
| Dirac weighting $K' = K - 1$ | 0.183/0.070 | 0.194/0.074 | 0.177/0.067 |
| Dirac weighting $K' = K - 2$ | 0.187/0.072 | 0.188/0.073 | 0.180/0.066 |
| Dirac weighting $K' = K - 4$ | 0.189/0.073 | 0.200/0.079 | 0.186/0.073 |

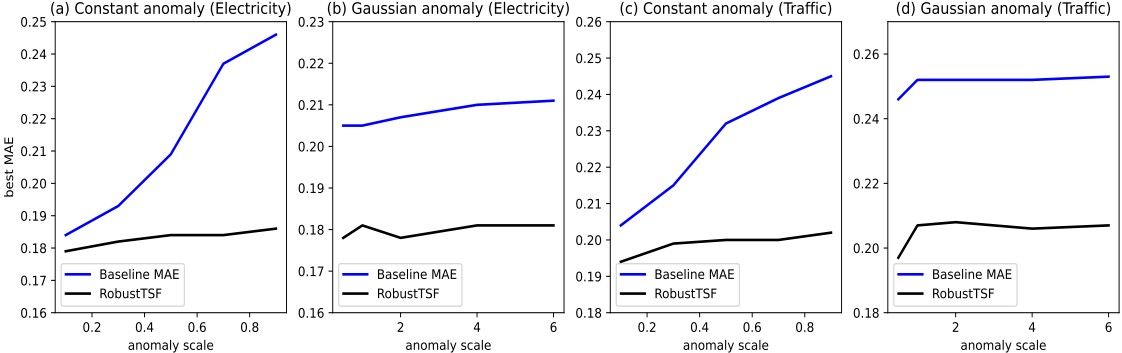

Figure 6: The figure depicts experiments on various anomaly scales, with a fixed anomaly ratio of 0.3 across all settings. The x-axis represents the anomaly scale and the y-axis represents the Best epoch MAE for each method. Notably, RobustTSF demonstrates robustness to different anomaly scales across diverse anomaly types. While the baseline also maintains robustness against Gaussian anomaly scales, its performance is impacted by Constant anomaly scales. It's worth mentioning that an anomaly scale experiment isn't feasible for the Missing type anomaly, as the anomaly scale remains consistently 0 due to the presence of missing values.

the x-axis corresponds to $\gamma^2$. Notably, the findings unveil that while the baseline (MAE) retains its robustness in the face of varying $\gamma^2$ (i.e., the variance of anomaly does not affect the methods very much), its performance does exhibit sensitivity to Constant anomaly scales.

### F.8 EFFICIENCY ANALYSES

We report training time for each method with LSTM structure on Electricity dataset with constant anomaly ($\eta = 0.3$) to compare the efficiency. The running hardware is MacBook with an Apple M1 chip.

The results are shown in Table 13. The small increased time of RobustTSF compared to Vanilla training is due to trend computation (Equation (4)). However, compared to detection-imputation-retraining pipeline, RobustTSF is much more efficient.

Table 13: Training time comparison of each method.

| Method | Vanilla Training | Offline imputation | Online imputation | RobustTSF |
|---|---|---|---|---|
| Time (s) | 27.41 | 68.09 | 89.33 | 36.84 |

### F.9 COMPARING ROBUSTTSF WITH MORE METHODS

There exist some methods, such as the recently proposed normalized Kalman filter (NKF) (de Bézenac et al., 2020), even not specifically designed for TSFA tasks, showing good performance when time series has missing data. Since NKF has better forecasting performance than DeepAR (Salinas et al.,

2020), KVAE (Krishnan et al., 2017), and GP-Copula (Salinas et al., 2019) on missing data settings, in this section, we mainly compare RobustTSF with the NKF method [3]. The results are shown in Figure 7, which shows that RobustTSF still exhibits better performance than NKF under different ratios of missing data.

We also compare RobustTSF with ARIMA, a popular non-DNN approach for time series forecasting. Specifically, we employed a rolling ARIMA model to suit one-step forward TSFA. Essentially, at each time step, we extended the noisy training time series by one step and used ARIMA to fit the new noisy training sequence, subsequently predicting the next step. However, ARIMA differs from DNN-based methods and cannot split time series. Comparing ARIMA and DNNs might introduce bias due to ARIMA's increased anomaly exposure. To mitigate this, we assume anomalies exist in both training and test. Appendix E1 provides details on applying RobustTSF to noisy test set. For ARIMA, we utilized the statsmodels package and finetuned (p,d,q) parameters (Due to ARIMA's characteristics, the validation time is long). Table 16 illustrates its performance falls short of RobustTSF due to its non-specialization for TSFA.

Another approach we compare is Dlinear (Zeng et al., 2023) which exhibits better performance than transformer-based models. The results are summarized in Table 14 (reporting best epoch MAE). The results demonstrate that, while DLinear may have a slight advantage in specific anomaly scenarios, RobustTSF consistently outperforms DLinear, particularly in Gaussian-type anomalies.

Table 14: Comparing RobustTSF with Dlinear. Reporting best epoch MAE for each method.

| Dataset | Method | Constant Anomaly | | Missing Anomaly | | Gaussian Anomaly | |
|---|---|---|---|---|---|---|---|
| | | $\eta = 0.1$ | $\eta = 0.3$ | $\eta = 0.1$ | $\eta = 0.3$ | $\eta = 0.1$ | $\eta = 0.3$ |
| Electricity | Dlinear | 0.176 | 0.189 | 0.179 | 0.198 | 0.195 | 0.347 |
| | RobustTSF | 0.177 | 0.183 | 0.181 | 0.194 | 0.176 | 0.177 |
| Traffic | Dlinear | 0.219 | 0.238 | 0.232 | 0.318 | 0.277 | 0.472 |
| | RobustTSF | 0.198 | 0.203 | 0.200 | 0.243 | 0.185 | 0.202 |

There also exists some potential approaches to deal with anomalies. For examples, after detecting anomalies, we can impute the values using interpolations or using the calculated trend as new time series to train the model. Table 15 shows the comparison with these methods on the Electricity dataset. The results show the superiority of RobustTSF over other methods.

Table 15: Comparing RobustTSF with interpolation-based and trend-only methods. The anomaly ratio for each anomaly type is 0.1 and 0.3, respectively. We report best epoch MAE for each method

| Method | Constant | | Missing | | Gaussian | |
|---|---|---|---|---|---|---|
| | $\eta = 0.1$ | $\eta = 0.3$ | $\eta = 0.1$ | $\eta = 0.3$ | $\eta = 0.1$ | $\eta = 0.3$ |
| zero-imputation | 0.191 | 0.212 | 0.200 | 0.224 | 0.190 | 0.232 |
| interpolate-imputation | 0.185 | 0.200 | 0.191 | 0.209 | 0.186 | 0.187 |
| Trend-only | 0.214 | 0.216 | 0.213 | 0.215 | 0.215 | 0.217 |
| RobustTSF | **0.182** | **0.190** | **0.187** | **0.204** | **0.182** | **0.177** |

Table 16: Comparing RobustTSF with non-DNN approach ARIMA. We report best epoch MAE for each method on Electricity dataset.

| Method | Gaussian Anomaly ($\eta = 0.1$) | Gaussian Anomaly ($\eta = 0.2$) | Gaussian Anomaly ($\eta = 0.3$) |
|---|---|---|---|
| ARIMA (1,0,0) | 0.213 | 0.294 | 0.419 |
| ARIMA (1,1,1) | 0.232 | 0.329 | 0.363 |
| ARIMA (5,1,0) | 0.230 | 0.327 | 0.405 |
| RobustTSF | **0.178** | **0.182** | **0.190** |

## F.10 APPLYING ROBUSTTSF ON OTHER ANOMALY DISTRIBUTIONS

In the main paper, we consider three types of anomalies, *i.e.,* Constant, Missing and Gaussian Anomaly. In this section, we consider more possible distributions for modeling anomalies, i.e., $z_t^A = z_t + \epsilon$, where $\epsilon$ follows heavy-tailed distributions as

---

[3]https://github.com/johannaSommer/KF_irreg_TS

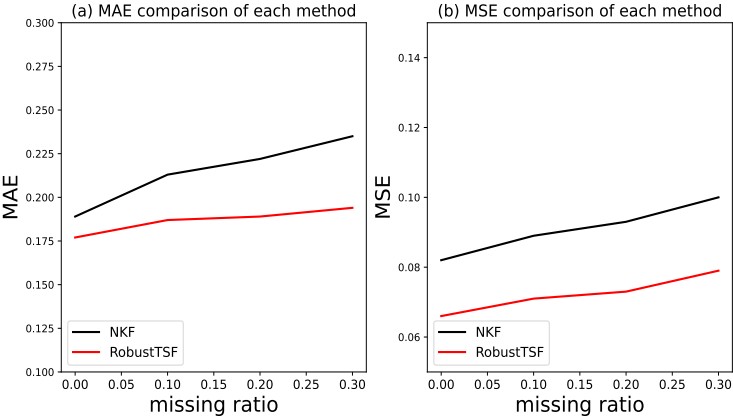

Figure 7: MAE and MSE comparison between RobustTSF and NKF (de Bézenac et al., 2020) on Electricity dataset with different ratios of missing data.

- Student-t distribution: $f(\epsilon, \nu) = \frac{\Gamma(\frac{\nu+1}{2})}{\sqrt{\nu\pi}\Gamma(\frac{\nu}{2})}(1 + \frac{\epsilon^2}{\nu})^{-\frac{(\nu+1)}{2}}$

- Generalized Pareto distribution: $f(\epsilon, c) = (1 + c \cdot \epsilon)^{-1-\frac{1}{c}}$

In the above, $f$ denotes pdf of the distributions and $\nu, c$ are hyper-parameters determining the shape of the distributions. We perform RobustTSF on these distributions and the results are shown in Table 17 and Table 18.

Table 17: Comparison of different methods on Electricity dataset for single-step forecasting with anomaly followed by student-t distribution. We report the best epoch performance for all methods, represented as MAE/MSE. The best results are highlighted in bold font.

| Method | Student-T ($\nu = 3$) | | | Student-T ($\nu = 6$) | | |
|---|---|---|---|---|---|---|
| | $\eta = 0.1$ | $\eta = 0.2$ | $\eta = 0.3$ | $\eta = 0.1$ | $\eta = 0.2$ | $\eta = 0.3$ |
| Vanilla (MAE) | 0.187/0.071 | 0.205/0.084 | 0.224/0.098 | 0.184/0.068 | 0.210/0.086 | 0.216/0.092 |
| Offline | 0.187/0.070 | 0.189/0.073 | 0.190/0.074 | 0.179/0.067 | 0.198/0.078 | 0.195/0.077 |
| Online | 0.196/0.075 | 0.196/0.074 | 0.217/0.089 | 0.197/0.075 | 0.200/0.081 | 0.219/0.091 |
| Loss sel | 0.190/0.073 | 0.198/0.078 | 0.208/0.087 | 0.194/0.076 | 0.203/0.083 | 0.203/0.084 |
| RobustTSF | **0.177/0.066** | **0.178/0.067** | **0.178/0.067** | **0.178/0.067** | **0.182/0.069** | **0.182/0.070** |

Table 18: Comparison of different methods on Electricity dataset for single-step forecasting with anomaly followed by generalized Pareto distribution. We report the best epoch performance for all methods, represented as MAE/MSE. The best results are highlighted in bold font.

| Method | Generalized-Pareto ($c = 1$) | | | Generalized-Pareto ($c = 0.5$) | | |
|---|---|---|---|---|---|---|
| | $\eta = 0.1$ | $\eta = 0.2$ | $\eta = 0.3$ | $\eta = 0.1$ | $\eta = 0.2$ | $\eta = 0.3$ |
| Vanilla (MAE) | 0.195/0.078 | 0.209/0.087 | 0.223/0.097 | 0.203/0.080 | 0.207/0.081 | 0.220/0.094 |
| Offline | 0.181/0.069 | 0.193/0.075 | 0.195/0.078 | 0.187/0.071 | 0.197/0.079 | 0.193/0.076 |
| Online | 0.192/0.073 | 0.201/0.078 | 0.209/0.086 | 0.193/0.074 | 0.207/0.083 | 0.212/0.086 |
| Loss sel | 0.195/0.076 | 0.204/0.084 | 0.205/0.085 | 0.191/0.074 | 0.194/0.076 | 0.208/0.086 |
| RobustTSF | **0.176/0.066** | **0.181/0.069** | **0.181/0.069** | **0.179/0.067** | **0.184/0.068** | **0.184/0.069** |

### F.11 APPLYING ROBUSTTSF ON MORE DATASETS

In this section, we evaluate RobustTSF on more datasets, including ETT (Zhou et al., 2021) and Exchange (Lai et al., 2018). The results are shown in Table 19, Table 20 and Table 21. Overall, RobustTSF still achieves better performance than other existing algorithms.

Table 19: Comparison of different methods on ETT and Exchange dataset. The model structure is LSTM. We report the best performance on test set, presented as MAE/MSE.

| Dataset | Method | Constant Anomaly | | Missing Anomaly | | Gaussian Anomaly | |
|---|---|---|---|---|---|---|---|
| | | $\eta = 0.1$ | $\eta = 0.3$ | $\eta = 0.1$ | $\eta = 0.3$ | $\eta = 0.1$ | $\eta = 0.3$ |
| ETT-h1 | Vanilla (MAE) | 0.054/0.006 | 0.061/0.007 | 0.053/0.005 | 0.070/0.008 | 0.053/0.006 | 0.055/0.006 |
| | Offline | 0.055/0.006 | 0.06/0.006 | 0.053/0.006 | 0.057/0.006 | 0.052/0.005 | 0.055/0.006 |
| | Online | 0.057/0.006 | 0.074/0.009 | 0.054/0.006 | 0.057/0.006 | 0.055/0.006 | 0.069/0.009 |
| | Loss sel | 0.055/0.006 | 0.062/0.007 | 0.055/0.006 | 0.065/0.009 | 0.052/0.006 | 0.060/0.007 |
| | RobustTSF | **0.052/0.005** | **0.054/0.005** | **0.052/0.005** | **0.055/0.006** | **0.052/0.005** | **0.055/0.006** |
| ETT-h2 | Vanilla (MAE) | 0.054/0.007 | 0.075/0.011 | 0.046/0.004 | 0.062/0.009 | 0.041/0.004 | 0.050/0.006 |
| | Offline | 0.041/0.003 | 0.074/0.012 | 0.045/0.005 | **0.060**/0.009 | 0.042/0.004 | 0.048/0.004 |
| | Online | 0.057/0.005 | 0.115/0.019 | 0.053/0.006 | 0.079/0.012 | 0.050/0.005 | 0.090/0.014 |
| | Loss sel | 0.042/0.003 | 0.080/0.014 | 0.047/0.005 | 0.061/0.009 | 0.044/0.004 | 0.057/0.007 |
| | RobustTSF | **0.040/0.003** | **0.058/0.007** | **0.041/0.003** | 0.061/**0.008** | **0.041/0.004** | **0.049/0.005** |
| ETT-m1 | Vanilla (MAE) | 0.026/0.001 | 0.029/0.002 | 0.026/0.001 | 0.028/0.002 | 0.026/0.001 | 0.027/0.002 |
| | Offline | 0.026/0.001 | 0.031/0.001 | 0.026/0.001 | 0.027/0.002 | 0.026/0.002 | 0.026/0.002 |
| | Online | 0.038/0.002 | 0.048/0.004 | 0.030/0.002 | 0.034/0.002 | 0.026/0.002 | 0.027/0.002 |
| | Loss sel | 0.026/0.001 | 0.038/0.003 | 0.026/0.001 | 0.031/0.002 | 0.026/0.002 | 0.027/0.002 |
| | RobustTSF | **0.025/0.001** | **0.026/0.001** | **0.026/0.001** | **0.026/0.001** | **0.025/0.001** | **0.026/0.001** |
| Exchange | Vanilla (MAE) | 0.123/0.029 | 0.131/0.029 | 0.160/0.050 | 0.140/0.039 | 0.124/0.032 | 0.208/0.085 |
| | Offline | 0.094/0.016 | 0.098/0.016 | 0.052/0.004 | 0.110/0.023 | 0.064/0.008 | 0.128/0.031 |
| | Online | **0.070/0.008** | **0.063/0.006** | 0.053/0.005 | 0.074/0.010 | 0.054/0.005 | 0.333/0.200 |
| | Loss sel | 0.121/0.027 | 0.228/0.095 | 0.130/0.030 | 0.229/0.101 | 0.141/0.043 | 0.253/0.124 |
| | RobustTSF | 0.072/0.010 | 0.076/0.011 | **0.052/0.004** | **0.057/0.006** | **0.053/0.005** | **0.065/0.008** |

Table 20: Comparison of different methods on ETT and Exchange dataset. The model structure is LSTM. We report the best performance on test set, presented as MAE/MSE.

| Dataset | Method | Student-T ($\nu = 3$) | | | Student-T ($\nu = 6$) | | |
|---|---|---|---|---|---|---|---|
| | | $\eta = 0.1$ | $\eta = 0.2$ | $\eta = 0.3$ | $\eta = 0.1$ | $\eta = 0.2$ | $\eta = 0.3$ |
| ETT-h1 | Vanilla (MAE) | 0.053/0.006 | 0.054/0.006 | 0.061/0.008 | 0.052/0.006 | 0.054/0.006 | 0.056/0.009 |
| | Offline | 0.053/0.005 | 0.055/0.006 | 0.062/0.007 | 0.053/0.005 | 0.054/0.006 | 0.058/0.007 |
| | Online | 0.056/0.006 | 0.062/0.007 | 0.065/0.008 | 0.058/0.007 | 0.060/0.007 | 0.065/0.008 |
| | Loss sel | 0.054/0.006 | 0.062/0.009 | 0.083/0.017 | 0.058/0.007 | 0.063/0.009 | 0.077/0.015 |
| | RobustTSF | **0.052/0.005** | **0.052/0.005** | **0.053/0.005** | **0.051/0.005** | **0.052/0.005** | **0.053/0.006** |
| ETT-h2 | Vanilla (MAE) | 0.042/0.004 | 0.048/0.005 | 0.052/0.006 | 0.045/0.004 | 0.046/0.005 | 0.057/0.007 |
| | Offline | 0.045/0.004 | **0.046**/0.005 | 0.053/0.006 | **0.042**/0.004 | 0.055/0.007 | 0.071/0.011 |
| | Online | 0.056/0.005 | 0.079/0.012 | 0.096/0.018 | 0.054/0.007 | 0.071/0.009 | 0.098/0.018 |
| | Loss sel | 0.042/0.004 | 0.055/0.006 | 0.061/0.009 | 0.042/0.004 | 0.052/0.006 | 0.058/0.008 |
| | RobustTSF | **0.042/0.004** | 0.048/**0.004** | **0.052/0.005** | 0.043/**0.004** | **0.045/0.005** | **0.050/0.005** |
| ETT-m1 | Vanilla (MAE) | 0.026/0.002 | 0.026/0.002 | 0.027/0.002 | 0.026/0.002 | 0.026/0.002 | 0.027/0.002 |
| | Offline | 0.026/0.002 | 0.026/0.002 | 0.026/0.002 | 0.026/0.002 | 0.026/0.002 | 0.026/0.002 |
| | Online | 0.026/0.002 | 0.026/0.002 | 0.026/0.002 | 0.026/0.002 | 0.026/0.002 | 0.026/0.002 |
| | Loss sel | 0.026/0.002 | 0.027/0.002 | 0.034/0.003 | 0.026/0.002 | 0.027/0.002 | 0.030/0.002 |
| | RobustTSF | **0.025/0.001** | **0.025/0.001** | **0.026/0.002** | **0.025/0.001** | **0.025/0.001** | **0.025/0.001** |
| Exchange | Vanilla (MAE) | 0.085/0.015 | 0.182/0.066 | 0.160/0.049 | 0.147/0.042 | 0.172/0.059 | 0.218/0.087 |
| | Offline | 0.082/0.013 | 0.102/0.019 | 0.137/0.036 | 0.087/0.014 | 0.106/0.021 | 0.133/0.033 |
| | Online | 0.053/0.004 | 0.104/0.021 | 0.203/0.075 | 0.058/0.005 | 0.095/0.016 | 0.092/**0.013** |
| | Loss sel | 0.174/0.063 | 0.198/0.082 | 0.254/0.125 | 0.146/0.042 | 0.153/0.046 | 0.321/0.091 |
| | RobustTSF | **0.046/0.003** | **0.051/0.004** | **0.081/0.012** | **0.060/0.006** | **0.061/0.007** | **0.083**/0.014 |

We also conduct experiments on time series datasets with real-world anomalies. The NAB benchmark dataset is from `http://odds.cs.stonybrook.edu/#table1.`. Table 23 shows the performace of RobustTSF is superior than other methods.

We provide a full description of the datasets we use in the paper in Table 22.

Table 21: Comparison of different methods on ETT and Exchange dataset. The model structure is LSTM. We report the best performance on test set, presented as MAE/MSE.

| Dataset | Method | Generalized-Pareto ($c = 1$) | | | Generalized-Pareto ($c = 0.5$) | | |
|---|---|---|---|---|---|---|---|
| | | $\eta = 0.1$ | $\eta = 0.2$ | $\eta = 0.3$ | $\eta = 0.1$ | $\eta = 0.2$ | $\eta = 0.3$ |
| ETT-h1 | Vanilla (MAE) | 0.052/0.006 | 0.054/0.006 | 0.057/0.006 | 0.053/0.005 | 0.053/0.006 | 0.064/0.008 |
| | Offline | 0.052/0.006 | 0.053/0.006 | **0.053**/0.006 | 0.052/0.005 | 0.053/0.005 | 0.056/0.006 |
| | Online | 0.054/0.006 | 0.055/0.006 | 0.061/0.006 | 0.055/0.006 | 0.058/0.006 | 0.064/0.007 |
| | Loss sel | 0.052/0.006 | 0.058/0.006 | 0.065/0.008 | 0.054/0.006 | 0.058/0.006 | 0.059/0.007 |
| | RobustTSF | **0.052/0.005** | **0.053/0.005** | 0.055/**0.006** | **0.052/0.005** | **0.052/0.005** | **0.056/0.006** |
| ETT-h2 | Vanilla (MAE) | 0.040/0.004 | 0.046/0.005 | 0.054/0.006 | 0.043/0.004 | 0.049/0.006 | 0.060/0.007 |
| | Offline | 0.041/0.004 | 0.044/0.005 | 0.049/0.005 | 0.046/0.004 | 0.046/0.004 | 0.051/0.005 |
| | Online | 0.049/0.005 | 0.055/0.007 | 0.075/0.010 | 0.049/0.004 | 0.062/0.006 | 0.075/0.009 |
| | Loss sel | 0.042/0.004 | 0.046/0.005 | 0.060/0.008 | 0.042/0.004 | 0.047/0.005 | 0.058/0.008 |
| | RobustTSF | **0.040/0.003** | **0.043/0.004** | **0.051/0.005** | **0.041/0.003** | **0.045/0.004** | **0.050/0.005** |
| ETT-m1 | Vanilla (MAE) | 0.025/0.002 | 0.026/0.002 | 0.028/0.002 | 0.026/0.002 | 0.026/0.002 | 0.027/0.002 |
| | Offline | 0.026/0.002 | 0.026/0.002 | 0.026/0.002 | 0.026/0.002 | 0.026/0.002 | 0.026/0.002 |
| | Online | 0.028/0.002 | 0.028/0.002 | 0.032/0.002 | 0.026/0.002 | 0.028/0.002 | 0.045/0.003 |
| | Loss sel | 0.026/0.002 | 0.026/0.002 | 0.026/0.002 | 0.026/0.002 | 0.026/0.002 | 0.029/0.002 |
| | RobustTSF | **0.026/0.001** | **0.026/0.002** | **0.026/0.002** | **0.026/0.001** | **0.026/0.001** | **0.026/0.002** |
| Exchange | Vanilla (MAE) | 0.090/0.019 | 0.104/0.023 | 0.077/0.012 | 0.070/0.007 | 0.112/0.024 | 0.091/0.015 |
| | Offline | 0.060/0.006 | 0.084/0.013 | 0.074/0.009 | 0.070/0.010 | 0.087/0.014 | 0.093/0.016 |
| | Online | 0.384/0.247 | 0.482/0.371 | 0.533/0.434 | 0.568/0.517 | 0.396/0.258 | 0.438/0.320 |
| | Loss sel | 0.100/0.024 | 0.144/0.040 | 0.141/0.038 | 0.105/0.020 | 0.137/0.037 | 0.175/0.054 |
| | RobustTSF | **0.060/0.007** | **0.064/0.008** | **0.071/0.009** | **0.044/0.003** | **0.064/0.008** | **0.078/0.012** |

Table 22: Description of the datasets used in the paper

| Dataset | LEN | FREQ | Manullay adding anomalies | Train Test ratio |
|---|---|---|---|---|
| Electricity | 26304 | 1h | Yes | 7:3 |
| Traffic | 17544 | 1h | Yes | 7:3 |
| ETT-h1 | 17420 | 1h | Yes | 7:3 |
| ETT-h2 | 17420 | 1h | Yes | 7:3 |
| ETT-m1 | 69680 | 15 min | Yes | 7:3 |
| Exchange | 7588 | 1 day | Yes | 7:3 |
| NAB (real-cloud) | 4032 | 15 min | Yes | 7:3 |
| M4-monthly | 1871 | monthly | Yes | 7:3 |

Table 23: Performance of each method on NAB benchmark (cloud dataset) and M4 forecasting dataset. Best epoch MAE is reported for each method. It can be observed that RobustTSF also performs well on real-world anomaly dataset and large forecasting dataset.

| | Vanilla MAE | Offline | Online | Loss sel | RobustTSF |
|---|---|---|---|---|---|
| NAB (real-cloud) | 1.28 | 1.20 | 1.14 | 2.11 | **0.87** |
| M4-monthly | 2.66 | 2.63 | 2.61 | 2.61 | **2.58** |
| M4-monthly (Const ano ratio 0.2) | 2.95 | 2.85 | 2.76 | 3.21 | **2.61** |

## F.12 Evaluating RobustTSF on Transformer and TCN Models

We use the transformer (Vaswani et al., 2017)[4] and temporal convolutional networks (TCN) (Bai et al., 2018)[5] to evaluate RobustTSF in single-step forecasting with different anomaly types on Electricity dataset. Results are shown in Table 24 and Table 25, respectively. It can be observed that RobustTSF also improves model performance on Transformer and TCN architectures.

---

[4]https://github.com/oliverguhr/transformer-time-series-prediction
[5]https://github.com/hyliush/deep-time-series/blob/master/models/TCN.py

Table 24: Evaluating RobustTSF on Transformer model for time series forecasting. We report the best epoch performance for the methods, represented as MAE/MSE. The anomaly ratio for each anomaly type is 0.1 and 0.3, respectively. The best results are highlighted in bold font.

| Method | Clean | Constant | | Missing | | Gaussian | |
|---|---|---|---|---|---|---|---|
| | $\eta = 0$ | $\eta = 0.1$ | $\eta = 0.3$ | $\eta = 0.1$ | $\eta = 0.3$ | $\eta = 0.1$ | $\eta = 0.3$ |
| Vanilla (MAE) | 0.266/0.139 | 0.271/0.139 | 0.304/0.184 | 0.287/0.152 | 0.385/0.352 | 0.263/0.141 | 0.302/0.188 |
| RobustTSF | **0.264/0.137** | **0.269/0.137** | **0.277/0.152** | **0.271/0.141** | **0.287/0.162** | **0.266/0.133** | **0.286/0.155** |

Table 25: Evaluating RobustTSF on TCN model for time series forecasting. We report the best epoch performance for the methods, represented as MAE/MSE. The anomaly ratio for each anomaly type is 0.1 and 0.3, respectively. The best results are highlighted in bold font.

| Method | Clean | Constant | | Missing | | Gaussian | |
|---|---|---|---|---|---|---|---|
| | $\eta = 0$ | $\eta = 0.1$ | $\eta = 0.3$ | $\eta = 0.1$ | $\eta = 0.3$ | $\eta = 0.1$ | $\eta = 0.3$ |
| Vanilla (MAE) | 0.186/0.083 | 0.199/0.083 | 0.218/0.093 | 0.205/0.088 | 0.232/0.103 | 0.190/**0.073** | 0.211/0.085 |
| RobustTSF | **0.185/0.078** | **0.189/0.080** | **0.192/0.081** | **0.186/0.078** | **0.206/0.087** | **0.185**/0.075 | **0.186/0.075** |

### F.13 Discussing the Originality, Significance and Future Work of RobustTSF

In this section, we discuss the originality, significance, and future work of RobustTSF, which collectively serve as a comprehensive conclusion to our paper.

**originality**: Our paper delves into the realm of Time Series Forecasting with Anomalies (TSFA), a relatively underexplored area within the time series domain. Traditional approaches to TSFA often employ the Detection-Imputation-Retraining (DIR) pipeline, which, while intuitive, lacks robust theoretical analyses and guarantees. In contrast, we draw inspiration from the advancements in Learning with Noisy Labels (LNL) and approach the TSFA task from a more principled perspective. We begin by providing a rigorous statistical definition of the TSFA problem. Leveraging our analyses on loss and sample robustness, we introduce an efficient filtering method called RobustTSF. This method stands in stark contrast to the DIR pipeline, delivering consistent and compelling empirical results across extensive experiments.

**significance**: Our analytical journey leads us to establish a notable connection between TSFA and Learning with Noisy Labels (LNL), a vibrant research direction in machine learning. Although applying LNL methods directly to TSFA is infeasible due to the regression nature of TSFA tasks and the presence of anomalies in the inputs of time series, our work demonstrates how to achieve robustness in TSFA tasks, aligning with the goals of LNL. This positions our proposed method, RobustTSF, with theoretical and empirical foundations. More significantly, it underscores the potential for drawing lessons from LNL to tackle TSFA tasks in the future.

Much like the evolution of LNL over recent years, progressing from modeling asymmetric/symmetric label noise to instance-dependent label noise, our modeling of time series anomalies and theoretical analyses can serve as a catalyst for future research. This can inspire the development of methods tailored to handle more intricate anomaly patterns in TSFA tasks.

**Future work**: Our primary focus centers on point-wise anomalies, with limited exploration of sequence-wise anomalies through specific experiments. The theoretical analysis of sequence-wise anomalies remains somewhat elusive. Real-world sequence anomalies can be intricate, and establishing statistical models and theoretical robustness against these anomalies poses a challenging aspect in our current work. While this topic falls outside the scope of this paper, it holds promise for future research endeavors.

**Limitation:** We present the limitation of the work below:

- Our method focuses on point-wise anomalies, with limited exploration of sequence-wise anomalies. The theoretical analysis of sequence-wise anomalies remains somewhat unclear, as real-world sequence anomalies can be complicated. The statistical modeling and theoretical robustness against these anomalies present a challenging and unexplored aspect.

- Our theoretical analysis is grounded in a statistical perspective, aligned with robust learning theory in the Learning with Noisy Labels (LNL) domain. However, in cases of limited training size, a potential disparity might arise between theoretical analyses and empirical findings.

From the limitation, the training size may impact the performance of RobustTSF, given that the theoretical analyses of losses are from a statistical perspective. Therefore, an implicit assumption of RobustTSF is that the training size should not be very small; otherwise, applying RobustTSF may not surpass baselines. While the datasets used in the paper suggest that training size may not be a significant issue, we conducted additional experiments to explore the gap between the baseline and RobustTSF on the Electricity dataset (reporting best epoch MAE for each setting) concerning the training size under a Gaussian anomaly setting (anomaly rate = 0.3). The results are presented in Table 26 which demonstrates that RobustTSF outperforms baselines when the training size of the dataset is not very small.

Table 26: Description of the datasets used in the paper

| fraction of original dataset | 0.5% | 1% | 2% | 10% | 50% | 100% |
|---|---|---|---|---|---|---|
| MAE | 1.08 | 0.901 | 0.713 | 0.578 | 0.240 | 0.210 |
| RobustTSF | 1.08 | 0.913 | 0.606 | 0.418 | 0.181 | 0.177 |

### F.14 EVALUATING ROBUSTTSF FOR LOW NOISE RATIOS

We conduct experiments of RobustTSF on Electricity dataset with low noise ratios and report the best epoch MAE for each setting. The results are reported in Table 27.

### F.15 ADDITIONAL ELUCIDATION FOR EQUATION (4)

To show that substitution of squared terms with absolute values in Equation (4) lead to an improvement in robustness. we use a simple time series function (sine function) to visualize the result of different options. Figure 8 shows that absolute term makes the trend more smooth and stable which would facilitate our anomaly score calculation in Equation (5).

### F.16 ADDITIONAL ABLATION STUDIES FOR ROBUSTTSF

We consider the effect of LNL and sample selection separately and conduct the ablation studies of RobustTSF on Electricity and Traffic datasets in Table 28

### F.17 PROOF OF STATISTICALLY CONSISTENT RESULTS FOR ROBUSTTSF

We reran the pivotal experiments outlined in Table 2 of our main paper, focusing on the Electricity dataset. We present the best epoch MAE as the mean (standard deviation) for the three runs of each configuration in Table 29. These results demonstrate the statistical consistency of our findings.

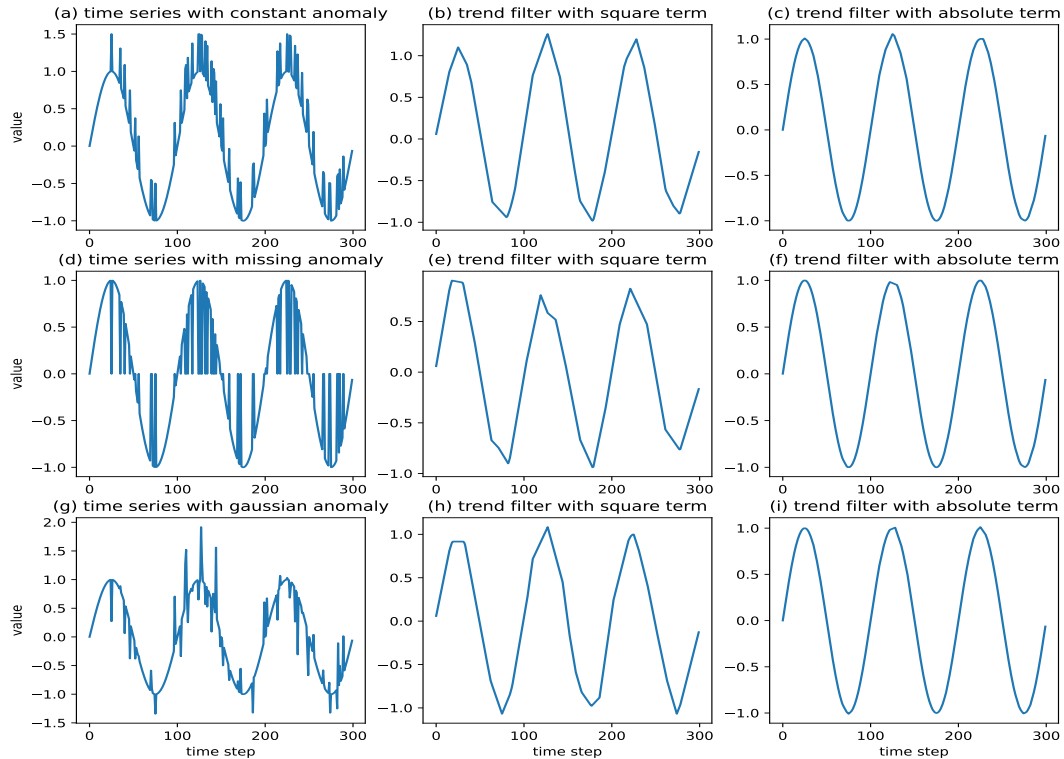

Figure 8: Illustration of the impact of square term and absolute term in trend filter.

Table 27: Comparing RobustTSF with other methods for low noise ratios. We report best epoch MAE for each method

| Method | Constant | | Missing | | Gaussian | |
|---|---|---|---|---|---|---|
| | $\eta = 0.01$ | $\eta = 0.05$ | $\eta = 0.01$ | $\eta = 0.05$ | $\eta = 0.01$ | $\eta = 0.05$ |
| MAE | 0.187 | 0.190 | 0.187 | 0.193 | 0.185 | 0.192 |
| MSE | 0.187 | 0.197 | 0.192 | 0.215 | 0.191 | 0.192 |
| Offline | 0.180 | 0.185 | 0.183 | 0.187 | 0.182 | 0.184 |
| Online | 0.187 | 0.193 | 0.188 | 0.190 | 0.188 | 0.190 |
| Loss sel | 0.192 | 0.195 | 0.192 | 0.196 | 0.187 | 0.190 |
| RobustTSF | **0.177** | **0.177** | **0.178** | **0.179** | **0.177** | **0.177** |

Table 28: Ablation studies of RobustTSF. Reporting best epoch MAE for each method.

| Dataset | Method | Constant Anomaly | | Missing Anomaly | | Gaussian Anomaly | |
|---|---|---|---|---|---|---|
| | | $\eta = 0.1$ | $\eta = 0.3$ | $\eta = 0.1$ | $\eta = 0.3$ | $\eta = 0.1$ | $\eta = 0.3$ |
| | Only LNL loss | 0.191 | 0.206 | 0.205 | 0.225 | 0.199 | 0.210 |
| Electricity | Only sample selection | 0.183 | 0.186 | 0.184 | 0.237 | 0.180 | 0.182 |
| | RobustTSF | 0.177 | 0.183 | 0.181 | 0.194 | 0.176 | 0.177 |
| | Only LNL loss | 0.209 | 0.231 | 0.266 | 0.295 | 0.216 | 0.233 |
| Traffic | Only sample selection | 0.199 | 0.220 | 0.224 | 0.310 | 0.196 | 0.210 |
| | RobustTSF | 0.198 | 0.203 | 0.200 | 0.243 | 0.185 | 0.202 |

Table 29: Comparing RobustTSF with other methods. We present the best epoch MAE as the mean (standard deviation) for the three runs of each configuration.

| Method | Clean | Constant | | Missing | | Gaussian | |
|---|---|---|---|---|---|---|---|
| | $\eta = 0$ | $\eta = 0.1$ | $\eta = 0.3$ | $\eta = 0.1$ | $\eta = 0.3$ | $\eta = 0.1$ | $\eta = 0.3$ |
| MSE | 0.187(0.002) | 0.200 (0.003) | 0.234 (0.003) | 0.227 (0.005) | 0.309 (0.015) | 0.185 | 0.192 (0.003) |
| MAE | 0.183(0.002) | 0.191(0.004) | 0.209 (0.004) | 0.207(0.003) | 0.223 (0.006) | 0.198 (0.003) | 0.210 (0.004) |
| Offline | 0.184(0.003) | 0.188 (0.003) | 0.217 (0.004) | 0.192 (0.003) | 0.213 (0.004) | 0.184 (0.002) | 0.191 (0.003) |
| Online | 0.184(0.002) | 0.199 (0.003) | 0.225 (0.003) | 0.194 (0.003) | 0.230 (0.003) | 0.195 (0.003) | 0.209 (0.002) |
| Loss sel | 0.180 (0.005) | 0.205(0.003) | 0.209 (0.004) | 0.201 (0.002) | 0.215(0.003) | 0.196 (0.002) | 0.201 (0.003) |
| RobustTSF | **0.177(0.002)** | **0.177 (0.002)** | **0.184 (0.002)** | **0.180(0.002)** | **0.195 (0.001)** | **0.175(0.001)** | **0.177 (0.001)** |

