# OpenReview forum: "RobustTSF: Towards Theory and Design of Robust Time Series Forecasting with Anomalies"
_ICLR.cc/2024/Conference — ICLR 2024 poster_

### Official Review · Reviewer_Swrw · 2023-10-27

**Soundness:** 2 fair
**Presentation:** 3 good
**Contribution:** 2 fair
**Rating:** 6
**Confidence:** 4

**Summary:**

The paper studies time series forecasting problem in the presence of anomalies, a setting where the difficulty comes from the fact that anomalies produce noise in the labels and in the input data at the same time. To deal with anomalies in the labels, authors propose to use Learn with Noisy Labels (LNL) framework and present analysis of LNL in the regression setting. They also introduce a heuristic approach to deal with anomalies in the input data by discarding samples where anomalies appear towards the end of an input sample. They combine both techniques in RobustTSF algorithm and compare it on two datasets with existing state-of-the-art approaches.

**Strengths:**

## Originality
* The analysis of mean absolute and mean square errors in LNL regression setting for 3 presented types of anomalies is novel
* The sample selection technique to avoid samples with late anomalies is novel
## Quality
* Authors compare their method to diverse set of baselines
* Empirical evaluation shows the advantage of presented techniques
## Clarity
Presentation is clear and main ideas are explained well
## Significance
I assess the paper to be of limited significance: while the presented method uses novel ideas in the context of time series forecasting with anomalies, evaluation is limited to two datasets with very similar properties and the limitations of the presented approach are not discussed.

**Weaknesses:**

* Hyper parameter tuning of RobustTSF is missing: it’s never discussed how \tau of 0.3 was chosen and Figure 5 suggests that choice of \tau would affect the comparison to the baselines.
* Overall limitations of the approach are not discussed. When does it break? What implicit assumptions are made? What time series aspect one would need to consider to choose this approach over another in practice?
* No confidence bounds in the experimental results makes it hard to judge the significance of the evaluation results
* Only two datasets of very similar properties are used for evaluation, which again limits the assessment of method applicability

Overall, my main concern is with the evaluation - choice of hyperparameters for the method and the lack of confidence bounds.

**Questions:**

* Proposition 2: the statement is not clear: does it hold under the conditions of Theorem 1? If yes, then the statement is obsolete, if not, then where does C_x comes from?
* Section 6.2: why notations change from z to x? It’s not clear what x is and how they are different from z
* End of page 6: while RobustTSD does not need a pre-trained DNN, it relies on limited trend estimation precedure in eq. (4) that implicitly assumes the smooth behavior of the time series. I’d see this as a disadvantage that one will have to adapt eq. (4) in unclear way to different datasets, while DNN would adopt automatically via pre-training.
* It’s great that authors look at scenario where the test set contains anomalies as well as this is the most realistic case. However, this sections misses a comparison of RobustTSF to other approaches on the test set with anomalies. It would be very relevant for practical applications if this evaluation is expanded.
* Ablation study section: what is presented is not an ablation study, but rather a hyperparameter tuning on the test dataset. An ablation study would, for example, consider the effect of LNL and sample selection separately, i.e. what happens when one applies only LNL or only sample selection.
* Parameter \tau seems to be scale dependent and while appendix mentions that the time series are normalized, the main text omits this detail.

---

> ### Author Response · Authors · 2023-11-21
> **Rebuttal-1**
>
> **W1:** evaluation is limited to two datasets with very similar properties.
>
> **A:** Our experiments encompassed eight datasets, featuring six instances with artificially introduced anomalies (detailed in the main paper and Appendix F.11) and two datasets with real-world anomalies (outlined in Appendix F.12). It's essential to note that, in contrast to image classification tasks where labels are often human-reviewed, time series datasets are typically recorded by sensors, and factors like temperature and humidity can introduce inherent noise. As illustrated in Table 2, RobustTSF demonstrated robust performance even without manually adding anomalies.
>
> -------------------------
>
> **W2:** Hyper parameter tuning of RobustTSF is missing: it’s never discussed how \tau of 0.3 was chosen and Figure 5 suggests that choice of \tau would affect the comparison to the baselines.
>
> **A:** The selection of $\tau$, denoting the filtering threshold akin to the threshold in loss-based sample selection for learning with noisy labels [R1], was fixed at 0.3 for all experiments. This choice was based on our observation that different values of τ around 0.3 did not significantly impact the performance of RobustTSF, as depicted in Figure 5. It's important to clarify that Figure 5 is not intended for comparing RobustTSF to other methods but rather to demonstrate the robustness of RobustTSF to variations in the hyperparameter $\tau$.
>
> [R1] co-teaching: robust training of deep neural networks with extremely noisy labels
>
> ------------------------------------------------
>
> **W3:** Overall limitations of the approach are not discussed. When does it break? What implicit assumptions are made? What time series aspect one would need to consider to choose this approach over another in practice?
>
> **A:** We provide the limitation of our method below and has added the content in our revised version.
>
> - Our method primarily focuses on point-wise anomalies, with limited exploration of sequence-wise anomalies through specific experiments. The theoretical analysis of sequence-wise anomalies remains somewhat unclear, as real-world sequence anomalies can be intricate. The statistical modeling and theoretical robustness against these anomalies present a challenging and unexplored aspect in our current work.
> - Our theoretical analysis is grounded in a statistical perspective, aligned with robust learning theory in the Learning with Noisy Labels (LNL) domain. However, in cases of limited training size, a potential disparity might arise between theoretical analyses and empirical findings.
>
> From the limitation, the training size may impact the performance of RobustTSF, given that the theoretical analyses of losses are from a statistical perspective. Therefore, an implicit assumption of RobustTSF is that the training size should not be very small; otherwise, applying RobustTSF may not surpass baselines. While the datasets used in the paper suggest that training size may not be a significant issue, we conducted additional experiments to explore the gap between the baseline and RobustTSF on the Electricity dataset (reporting best epoch MAE for each setting) concerning the training size under a Gaussian anomaly setting (anomaly rate = 0.3):
>
>
> |           | 0.5% | 1%    | 2%    | 10%   | 50%   | 100%  |
> | --------- | ---- | ----- | ----- | ----- | ----- | ----- |
> | MAE       | 1.08 | 0.901 | 0.713 | 0.578 | 0.240 | 0.210 |
> | RobustTSF | 1.08 | 0.913 | 0.606 | 0.418 | 0.181 | 0.177 |
>
> The results demonstrate that RobustTSF outperforms baselines when the training size of the dataset is not very small.

---

> ### Author Response · Authors · 2023-11-21
> **Rebuttal-2**
>
> **W4:** No confidence bounds in the experimental results makes it hard to judge the significance of the evaluation results
>
> **A:** To show our results are statistically consistent,Specifically, we reran the pivotal experiments outlined in Table 2 of our main paper, focusing on the Electricity dataset. We present the best epoch MAE as the mean (standard deviation) for the three runs of each configuration, as follows:
>
>
> | Electricity | Clean         | Const 0.1     | const 0.3     | Missing 0.1   | Missing 0.3   | Gaussian 0.1  | Gaussian 0.3  |
> | ----------- | ------------- | ------------- | ------------- | ------------- | ------------- | ------------- | ------------- |
> | MSE         | 0.187(0.002)  | 0.200 (0.003) | 0.234 (0.003) | 0.227 (0.005) | 0.309 (0.015) | 0.192 (0.003) | 0.228 (0.002) |
> | MAE         | 0.183(0.002)  | 0.191(0.004)  | 0.209 (0.004) | 0.207 0.003() | 0.223 (0.006) | 0.198 (0.003) | 0.210 (0.004) |
> | Offline     | 0.184(0.003)  | 0.188 (0.003) | 0.217 (0.004) | 0.192 (0.003) | 0.213 (0.004) | 0.184 (0.002) | 0.191 (0.003) |
> | Online      | 0.184(0.002)  | 0.199 (0.003) | 0.225 (0.003) | 0.194 (0.003) | 0.230 (0.003) | 0.195 (0.003) | 0.209 (0.002) |
> | Loss sel    | 0.180 (0.005) | 0.205(0.003)  | 0.209 (0.004) | 0.201 (0.002) | 0.215(0.003)  | 0.196 (0.002) | 0.201 (0.003) |
> | RobustTSF   | 0.177(0.002)  | 0.177 (0.002) | 0.184 (0.002) | 0.180(0.002)  | 0.195(0.001)  | 0.175(0.001)  | 0.177 (0.001) |
>
>
> These results demonstrate the statistical consistency of our findings.
>
> ------------------------
>
> **Q1:** Proposition 2: the statement is not clear: does it hold under the conditions of Theorem 1? If yes, then the statement is obsolete, if not, then where does C_x comes from?
>
> **A:** The definition of $C_x$ in both Theorem 1 and Proposition 2 remains consistent. However, it's crucial to note that these propositions address different scenarios. The condition outlined in Theorem 1 pertains to the assumption that $C_x$ is constant with respect to the  forecasting models. In Proposition 2, we explore loss functions where $C_x$ does not remain constant with respect to the  forecasting models. Specifically, Proposition 2 aims to demonstrate that if $C_x$ can be estimated, it allows us to normalize the loss, thereby enhancing its robustness. This insight draws inspiration from the field of learning with noisy labels in (image) classification, where techniques such as normalizing cross entropy have been employed to enhance robustness to label noise [R1].
>
> [R1] normalized loss functions for deep learning with noisy labels
>
> ------------------------
>
> **Q2:** Section 6.2: why notations change from z to x? It’s not clear what x is and how they are different from z
>
> **A:** As explained in our problem formulation in Section 3, for a given time series with anomalies, $\widetilde{z_i}$ is the i-th value and $\widetilde{\mathbf{x}}\_{i}$ is the i-th segment. The length of $\widetilde{\mathbf{x}}\_{i}$ is $\alpha$. This notation is consistently applied in Section 6.2.
>
> ------------------------
>
> **Q3:** End of page 6: while RobustTSD does not need a pre-trained DNN, it relies on limited trend estimation precedure in eq. (4) that implicitly assumes the smooth behavior of the time series. I’d see this as a disadvantage that one will have to adapt eq. (4) in unclear way to different datasets, while DNN would adopt automatically via pre-training.
>
> **A:** We acknowledge the need for trend estimation in RobustTSF, and we agree that this estimation may not be accurate for every dataset. The main objective of our paper is to highlight the time efficiency of RobustTSF in comparison to DIR (detection imputation retraining pipeline), which relies on pre-trained DNNs. As demonstrated in [R1], DIR involves multiple steps, including training a DNN on noisy time series and then using this pre-trained DNN for anomaly detection. However, this multi-step process increases training time and may result in an inferior pre-trained DNN, especially when trained on noisy data.
>
> [R1] Recurrent neural networks and robust time series prediction

---

> ### Author Response · Authors · 2023-11-21
> **Rebuttal-3**
>
> **Q4:** It’s great that authors look at scenario where the test set contains anomalies as well as this is the most realistic case. However, this sections misses a comparison of RobustTSF to other approaches on the test set with anomalies. It would be very relevant for practical applications if this evaluation is expanded.
>
> **A:** Thank you for your suggestion. We have expanded the evaluation to include scenarios where the test set contains anomalies. The comparative results with other methods are presented in the table below:
>
> | Electricity | Const 0.1 | const 0.3 | Missing 0.1 | Missing 0.3 | Gaussian 0.1 | Gaussian 0.3 |
> | ----------- | --------- | --------- | ----------- | ----------- | ------------ | ------------ |
> | MAE         | 0.192     | 0.253     | 0.201       | 0.313       | 0.200        | 0.258        |
> | MSE         | 0.211     | 0.281     | 0.237       | 0.387       | 0.212        | 0.275        |
> | Offline     | 0.190     | 0.232     | 0.198       | 0.324       | 0.191        | 0.255        |
> | Online      | 0.197     | 0.270     | 0.202       | 0.362       | 0.212        | 0.250        |
> | Loss sel    | 0.207     | 0.235     | 0.208       | 0.311       | 0.199        | 0.236        |
> | RobustTSF   | 0.174     | 0.219     | 0.183       | 0.302       | 0.178        | 0.190        |
>
> These results demonstrate that RobustTSF outperforms other methods on test sets with anomalies, providing valuable insights for practical applications.
>
> ---------------------------
>
> **Q5:** Ablation study section: what is presented is not an ablation study, but rather a hyperparameter tuning on the test dataset. An ablation study would, for example, consider the effect of LNL and sample selection separately, i.e. what happens when one applies only LNL or only sample selection.
>
> **A:** Thank you for your feedback. We have updated the section title in the main paper to "Hyperparameter Tuning" and conducted the suggested ablation studies. The results are presented in the table below:
>
> | Electricity           | Const 0.1 | const 0.3 | Missing 0.1 | Missing 0.3 | Gaussian 0.1 | Gaussian 0.3 |
> | --------------------- | --------- | --------- | ----------- | ----------- | ------------ | ------------ |
> | Only LNL (loss)       | 0.191     | 0.206     | 0.205       | 0.225       | 0.199        | 0.210        |
> | Only sample selection | 0.183     | 0.186     | 0.184       | 0.237       | 0.180        | 0.182        |
> | RobustTSF             | 0.177     | 0.183     | 0.181       | 0.194       | 0.176        | 0.177        |
>
> | Traffic           | Const 0.1 | const 0.3 | Missing 0.1 | Missing 0.3 | Gaussian 0.1 | Gaussian 0.3 |
> | --------------------- | --------- | --------- | ----------- | ----------- | ------------ | ------------ |
> | Only LNL (loss)       | 0.209     | 0.231     | 0.266       | 0.295       | 0.216        | 0.233        |
> | Only sample selection | 0.199     | 0.220     | 0.224       | 0.310       | 0.196        | 0.210        |
> | RobustTSF             | 0.198     | 0.203     | 0.200       | 0.243       | 0.185        | 0.202        |
>
> These additional analyses have been included in the revised version of our paper.
>
> -----------------------------------
>
> **Q6:** Parameter \tau seems to be scale dependent and while appendix mentions that the time series are normalized, the main text omits this detail.
>
> **A:** Thank you for pointing out the omission. We have addressed this by adding a clarification in the main paper. Specifically, the data is normalized to have a mean of 0 and a standard deviation of 1 before adding anomalies and training process. Additionally, it's worth noting that we consistently set the parameter $\tau$ to 0.3 for all configurations across different datasets. The normalized information has been included in the revised version of our paper.

---

> > ### Author Response · Authors · 2023-11-21
> > **Response to Reviewer Swrw for discussion**
> >
> > Dear Reviewer Swrw,
> >
> > Since the End of author/reviewer discussions is coming soon and we also tried our best to address your questions, may we know if our response addresses your main concerns? If so, we kindly ask for your reconsideration of the score. Should you have any further advice on the paper and/or our rebuttal, please let us know and we will be more than happy to engage in more discussion and paper improvements.
> >
> > Thank you so much for devoting time to improving our paper!

---

> ### Comment · Reviewer_Swrw · 2023-11-22
> **Acknowledgement of authors' response**
>
> Thank you for the clarifying details. You addressed my main concern regarding confidence bounds in the experiments (even though I would expect this to be included in Table 2 as important data). Hence I am inclined to raise my score to 6.

---

> > ### Author Response · Authors · 2023-11-22
> >
> > We express our gratitude to the reviewer for supporting our paper and providing constructive comments to enhance its quality. We acknowledge the importance of the results presented. As Table 2 is already densely populated with both MAE and MSE evaluations, we have included these additional results in the Appendix, with a reference added in the caption of Table 2 in the main paper.

---

### Official Review · Reviewer_442h · 2023-10-31

**Soundness:** 3 good
**Presentation:** 2 fair
**Contribution:** 2 fair
**Rating:** 5
**Confidence:** 4

**Summary:**

The paper introduces a method that combines the learning with noisy labels (LNL) technique and time series forecasting with anomalies (TSFA). The key distinction is that LNL deals with label noise in the target variable Y, while TSFA handles noise in both the input variable X and the target variable Y, making TSFA more complex than LNL. The authors categorize anomalies into three types: constant, missing, and Gaussian type anomalies. They analyze the loss robustness when anomalies affect Y and the sample robustness when anomalies affect X.
The authors propose a method called RobustTSF that integrates LNL and TSFA. Specifically, they provide a method for selecting time series with minimal anomalies. The process involves computing the trend of a given time series, denoted as S, and constructing a triplet D = (X, S, Y). An anomaly score A is defined as the difference between X and S, with an associated weight w. After calculating the anomaly scores for all samples, the authors design the final loss function for RobustTSF, employing the mean absolute error (MAE) as a more robust loss function. They utilize the anomaly score to measure the degree of anomalies in the time series data and perform appropriate filtering.
Importantly, this method does not rely on any specific deep learning model and can be compatible with any DNN model for time series forecasting.

**Strengths:**

1.Originality: This article presents a novel method that integrates LNL and TSFA, representing the first application of LNL to the domain of time series prediction. This pioneering approach demonstrates significant originality and provides valuable guidance for future research directions.
2.Quality: This article exhibits high quality as it presents a well-supported argument, effectively connecting LNL and TSFA from a new perspective, allowing readers to comprehend the rationale behind the author's design choices. Furthermore, the author validates the effectiveness of this approach through various experiments, enhancing the overall quality of the work.
3.Clarity: The author demonstrates exceptional clarity through a meticulously organization, guiding readers step by step in understanding the algorithm. Moreover, the author employs precise terminology, minimizing ambiguity and ensuring a clear and unambiguous understanding of the concepts presented.
4.Significance: The author's work holds great significance as they propose a novel framework for addressing anomalies in time series data, applicable to any forecasting model. This contribution provides a fresh direction for researchers in the field of time series prediction, offering an alternative approach to further enhance prediction accuracy. The implications of this work are of paramount importance to the community, as it opens up new avenues for exploration and advancement in the field.

**Weaknesses:**

1. I noticed that in your experimental section, you used a large input length to predict a relatively small output length. For example, in section 7.2, you used an input length of 96 to predict an output data of length [4, 8]. However, for time series forecasting tasks, the typical setting is an input length of 12 and an output length of 12. Therefore, I am unsure if your model is applicable to this setting. I recommend conducting an additional set of experiments where the input series has a length of 12 and the output series has a length of 12, in order to assess the performance of your model in this specific scenario.
2. In Section 6.2, I propose that additional elucidation should be provided: Why does the substitution of squared terms with absolute values in Equation (4) lead to an improvement in robustness?
3. Please note the spelling error: "k-the" below Equation (5) should be corrected to "k-th."

**Questions:**

1.In Theorem 1, why is it necessary for the anomaly rate to be less than 0.5?
2.In your method, you mentioned three types of anomalies, and your analysis of loss robustness indicates that MAE is robust to all three types of anomalies, while MSE is only robust to one of them. Can we understand from this that MAE is more suitable as a loss function than MSE in all situations? Are there any criteria to evaluate the robustness of other loss functions?

---

> ### Author Response · Authors · 2023-11-21
> **Rebuttal**
>
> **W1:** I noticed that in your experimental section, you used a large input length to predict a relatively small output length. For example, in section 7.2, you used an input length of 96 to predict an output data of length [4, 8]. However, for time series forecasting tasks, the typical setting is an input length of 12 and an output length of 12. Therefore, I am unsure if your model is applicable to this setting. I recommend conducting an additional set of experiments where the input series has a length of 12 and the output series has a length of 12, in order to assess the performance of your model in this specific scenario.
>
> **A:** We appreciate your observation regarding the input and output lengths in our experiments. Following your suggestion, we conducted additional experiments on the Electricity dataset with an input length of 12 and an output length of 12 and report the best epoch MAE. The results are presented below:
>
> |           | Const 0.1 | Const 0.3 | Missing 0.1 | Missing 0.3 | Gaussian 0.1 | Gaussian 0.3 |
> | --------- | --------- | --------- | ----------- | ----------- | ------------ | ------------ |
> | MAE       | 0.550     | 0.575     | 0.544       | 0.572       | 0.541        | 0.565        |
> | MSE       | 0.551     | 0.578     | 0.545       | 0.595       | 0.545        | 0.568        |
> | Offline   | 0.539     | 0.552     | 0.527       | 0.616       | 0.531        | 0.546        |
> | Online    | 0.623     | 0.627     | 0.617       | 0.680       | 0.605        | 0.654        |
> | Loss sel  | 0.560     | 0.564     | 0.535       | 0.623       | 0.549        | 0.563        |
> | RobustTSF | 0.512     | 0.543     | 0.525       | 0.547       | 0.516        | 0.523        |
>
> These results affirm that RobustTSF maintains good performance with input and output length of 12. We have included this new set of experiments in our revised version, and the corresponding code is provided in the supplementary material, named “RobustTSF_multi_step”.
>
> ---------------------------
>
> **W2:** In Section 6.2, I propose that additional elucidation should be provided: Why does the substitution of squared terms with absolute values in Equation (4) lead to an improvement in robustness?
>
> **A:** Experimentally, we observed that substituting squared terms with absolute values enhances the stability of the model's performance. We agree with the reviewer that additional elucidation is needed. In Appendix F.15, we use a simple time series function (sine function) to visualize the result of different options. It shows that absolute term makes the trend more smooth and stable, which would facilitate our anomaly score calculation in Equation (5).
>
> ---------------------------
>
> **W3:** Please note the spelling error: "k-the" below Equation (5) should be corrected to "k-th."
>
> **A:** Thank you for pointing out the spelling error. We have corrected "k-the" to "k-th".
>
> ---------------------------
>
> **Q:** In Theorem 1, why is it necessary for the anomaly rate to be less than 0.5? 2.In your method, you mentioned three types of anomalies, and your analysis of loss robustness indicates that MAE is robust to all three types of anomalies, while MSE is only robust to one of them. Can we understand from this that MAE is more suitable as a loss function than MSE in all situations? Are there any criteria to evaluate the robustness of other loss functions?
>
> **A:** Regarding Theorem 1, the necessity for the anomaly rate to be less than 0.5 is derived from Appendix A.1, where the proof indicates that to align the loss on noisy time series labels consistently with clean labels, $1-2\eta$ must be greater than 0. Therefore, the anomaly ratio must be less than 0.5.
>
> Concerning the choice between MAE and MSE, our analysis indicates that MAE is more suitable, as it exhibits robustness to all three types of anomalies. Previous TSFA research mainly use MSE and our study shows that MSE is only robust to Gaussian anomalies.
> We acknowledge your observation and provide Proposition 2 as a criterion to evaluate other potential loss functions. If a loss function satisfies the conditions outlined in Proposition 2, it can be considered robust to Constant and Missing type anomalies. However, the exploration of additional criteria for the theoretical evaluation of alternative loss functions is outside the scope of this paper, and we consider it as a potential avenue for future research.

---

> > ### Author Response · Authors · 2023-11-21
> > **Response to Reviewer 442h (before the end of rebuttal)**
> >
> > Dear Reviewer 442h,
> >
> > Thank you so much for devoting time to improving our work!
> >
> > Since the End of author/reviewer discussions is just in one day, may we know if our response addresses your main concerns? If so, we kindly ask for your reconsideration of the score. Should you have any further advice on the paper and/or our rebuttal, please let us know and we will be more than happy to engage in more discussion and paper improvements.

---

> > > ### Comment · Reviewer_442h · 2023-11-22
> > > **Acknowledgement of authors' response**
> > >
> > > Thank the authors for the response. After carefully reading all other reviewers' comments and rebuttals, I would like to keep my original score.

---

### Official Review · Reviewer_FUy4 · 2023-11-02

**Soundness:** 2 fair
**Presentation:** 2 fair
**Contribution:** 2 fair
**Rating:** 5
**Confidence:** 3

**Summary:**

This article presents a model for forecasting time series containing anomalies. The article begins by specifying the three main types of anomalies of interest and then introduces the theorems that will be used for the proposed algorithm. These theorems are primarily proofs of the robustness of Mean Squared Error (MSE) and Mean Absolute Error (MAE) in the presence of anomalies. The proposed algorithm mainly involves filtering to detect potential anomalies and penalizing the cost for windows containing these anomalies in order to not consider them. Experiments are finally conducted to demonstrate the method's efficiency in various settings

**Strengths:**

* The state of the art is well presented, the work presented is well contextualized, and the theoretical aspects are well introduced and explained
* The experiments are numerous, they are entirely reproducible, and the appendices contain very useful information.
* The issue of anomalies is well introduced, formally presented correctly, and intriguingly

**Weaknesses:**

* The first weakness of the article is the presentation of the main algorithm's idea. Its explanation is spread across several pages, and one has to wait until the 6th page to get an idea of how the algorithm will work, even though the idea itself is rather simple: identifying anomalies in the signal and applying very weak penalties to the cost in the corresponding region. It would be much easier to read the article if even a brief description were introduced very early on to understand the relevance of the different theorems formulated.

* Regarding novelty, the main theorems seem to be adapted quite directly from other works, so there isn't much novelty there. Furthermore, the idea of filtering and training on robust windows is not very new either, and the filtering method is also pre-existing. Apart from adapting the framework for anomalies to time series, there isn't a significant novelty.

* Regarding Section 5, which analyzes the effect of the position of anomalies on forecasting, it seems quite normal that anomalies near the end of the signal disrupt the prediction more than anomalies at the beginning. The system's state is disrupted, making forecasting more challenging. In contrast, if the anomalies are at the beginning, the model has time to stabilize on the 'correct' values.

* However, the biggest weakness of the article lies in the experiments, which do not include any modern time series forecasting baselines (such as PatchTST, DLinear, Autoformer, ...). From a few quick experiments I conducted using the authors' code, DLinear seems to easily outperform the proposed model on the noisy data. This raises the question of the method's relevance if state-of-the-art models do not struggle with the anomalies considered by the authors. Of course, I may be mistaken, but in any case, it would be helpful to have the performance of state-of-the-art algorithms on this kind of problem to understand the challenges posed by the envisaged anomalies.

**Questions:**

Can you provide experiments with the usual baseline of the state of the art ? (PatchTST, Dlinear, Autoformer, Pyformer, ...)

---

> ### Author Response · Authors · 2023-11-21
> **Rebuttal-1**
>
> **W1**: The first weakness of the article is the presentation of the main algorithm's idea. Its explanation is spread across several pages, and one has to wait until the 6th page to get an idea of how the algorithm will work, even though the idea itself is rather simple: identifying anomalies in the signal and applying very weak penalties to the cost in the corresponding region. It would be much easier to read the article if even a brief description were introduced very early on to understand the relevance of the different theorems formulated.
>
> **A:** We apologize for any confusion caused by the current organization. The structure aligns with the novelty of the task in time series forecasting with anomalies (TSFA), where formal definitions of problem formulations and anomaly types are crucial. Sections 4 and 5 are dedicated to analyzing the distinctions between TSFA and learning with noisy labels (LNL). We acknowledge the need for an early algorithm description and have addressed this by adding it to the Introduction section.
>
> ---------------------
>
> **W2:** Regarding novelty, the main theorems seem to be adapted quite directly from other works, so there isn't much novelty there. Furthermore, the idea of filtering and training on robust windows is not very new either, and the filtering method is also pre-existing. Apart from adapting the framework for anomalies to time series, there isn't a significant novelty.
>
> **A:**  We clarify that the original robust loss theorem we followed was designed for classification in conventional LNL settings [R1]. TSFA introduces a regression problem with unique anomalies, necessitating the formal definition and examination of loss robustness specific to TSFA. To our knowledge, no previous studies address loss robustness in TSFA tasks.
>
> While the concept of filtering has been explored in earlier TSFA studies [R2], it is essential to recognize notable distinctions. Previous filtering strategies, as observed in [R2], apply the same approach as in conventional LNL problems, without taking into account the nuances specific to TSFA. In contrast, our filtering strategy is meticulously crafted by considering the disparities between TSFA and LNL, resulting in consistent performance improvement across diverse anomaly types.
>
> [R1] Robust loss functions under label noise for deep neural networks.
>
> [R2] Robust learning of deep time series anomaly detection models with contaminated training data.
>
> -------------------------
>
>
> **W3:** Regarding Section 5, which analyzes the effect of the position of anomalies on forecasting, it seems quite normal that anomalies near the end of the signal disrupt the prediction more than anomalies at the beginning. The system's state is disrupted, making forecasting more challenging. In contrast, if the anomalies are at the beginning, the model has time to stabilize on the 'correct' values.
>
> **A:**  Although the observation may appear commonplace, it serves as a foundational point of distinction for TSFA compared to LNL (Learning with Noisy Labels). This observation directs the algorithm design, enabling the selection of input time series where anomalies are sparse in the time series points located near the label. Leveraging sample robustness and loss robustness, the algorithm achieves robustness goals akin to general LNL tasks. Similar findings have been noted in temporal graph learning [R1], emphasizing the significance of recent edges for target predictions.
>
> [R1] Adaptive data augmentation on temporal graph

---

> ### Author Response · Authors · 2023-11-21
> **Rebuttal-2**
>
> **W4&Q1:** However, the biggest weakness of the article lies in the experiments, which do not include any modern time series forecasting baselines (such as PatchTST, DLinear, Autoformer, ...). From a few quick experiments I conducted using the authors' code, DLinear seems to easily outperform the proposed model on the noisy data. This raises the question of the method's relevance if state-of-the-art models do not struggle with the anomalies considered by the authors. Of course, I may be mistaken, but in any case, it would be helpful to have the performance of state-of-the-art algorithms on this kind of problem to understand the challenges posed by the envisaged anomalies. & Can you provide experiments with the usual baseline of the state of the art ? (PatchTST, Dlinear, Autoformer, Pyformer, ...)
>
> **A:** Firstly, in the context of learning with noisy label tasks (LNL), it is conventional to maintain a consistent network structure for all methods to ensure a fair evaluation of robustness and superiority [R1, R2, R3] (eliminating network structure bias). Our paper proposes a model-agnostic method, necessitating the use of the same network structure for all methods. LSTM is chosen due to its prevalence in recent robust time series forecasting papers [R4, R5, R6], as well as in related works on TSFA tasks [R7, R8, R9].
>
> While we acknowledge the importance of exploring other network structures, we demonstrate the versatility of RobustTSF in Appendix F.12 by showcasing improved performance with TCN and transformer models. Notably, PatchTST, Autoformer, and Pyformer are transformer-based models.
>
> Secondly, concerning DLinear [R10], which is asserted to have superior performance to transformer-based models, we have thoroughly analyzed and experimented with it. Analytically, DLinear's reliance on linear regression for both trend and seasonality may limit its performance since the randomness of anomalies introduces challenges in trusting the seasonality of input time series. Moreover, Table 15 in our paper illustrates that relying solely on the trend yields inferior performance compared to RobustTSF.
>
> Experimentally, we ran DLinear on the Electricity and Traffic datasets using the official code [R11], incorporating various anomaly types as detailed in Table 2. The results are summarized below (reporting best epoch MAE):
>
> | Dataset: Electricity | Const 0.1 | Const 0.3 | Missing 0.1 | Missing 0.3 | Gaussian 0.1 | Gaussian 0.3 |
> | -------------------- | --------- | --------- | ----------- | ----------- | ------------ | ------------ |
> | Dlinear              | 0.176     | 0.189     | 0.179       | 0.198       | 0.195        | 0.347        |
> | RobustTSF            | 0.177     | 0.183     | 0.181       | 0.194       | 0.176        | 0.177        |
>
>
> | Dataset: Traffic | Const 0.1 | Const 0.3 | Missing 0.1 | Missing 0.3 | Gaussian 0.1 | Gaussian 0.3 |
> | ---------------- | --------- | --------- | ----------- | ----------- | ------------ | ------------ |
> | Dlinear          | 0.219     | 0.238     | 0.232       | 0.318       | 0.277        | 0.472        |
> | RobustTSF        | 0.198     | 0.203     | 0.200       | 0.243       | 0.185        | 0.202
>
> These results illustrate that, although DLinear may exhibit a slight advantage in certain anomaly scenarios, RobustTSF consistently outperforms DLinear in the majority of settings, especially when confronted with Gaussian-type anomalies. The code for DLinear is available in the supplementary material under the filename "train_noisy_dlinear.py."
>
>
> [R1] Co-teaching: Robust Training of Deep Neural Networks with Extremely Noisy Labels
>
> [R2] Generalized cross entropy loss for training deep neural networks with noisy labels
>
> [R3] Early-Learning Regularization Prevents Memorization of Noisy Labels
>
> [R4]  Robust probabilistic time series forecasting.
>
> [R5] Huber additive models for non-stationary time series analysis.
>
> [R6] Dynamic gaussian mixture based deep generative model for robust forecasting on sparse multivariate time series.
>
> [R7] Recurrent neural networks and robust time series prediction
>
> [R8] Resilient neural forecasting systems
>
> [R9] Robust learning of deep time series anomaly detection models with contaminated training data.
>
> [R10] Are Transformers Effective for Time Series Forecasting?
>
> [R11] https://github.com/cure-lab/LTSF-Linear/blob/main/models/DLinear.py

---

> > ### Author Response · Authors · 2023-11-21
> > **Response to Reviewer FUy4 before the end of discussion**
> >
> > Dear Reviewer FUy4,
> >
> > Since the End of author/reviewer discussions is just in one day, may we know if our response addresses your main concerns? If so, we kindly ask for your reconsideration of the score.
> >
> > Should you have any further advice on the paper and/or our rebuttal, please let us know and we will be more than happy to engage in more discussion and paper improvements. We would really appreciate it if our next round of communication could leave time for us to resolve any of your remaining or new questions.
> >
> > Thank you so much for devoting time to improving our work!

---

> > ### Comment · Reviewer_FUy4 · 2023-11-22
> > **Acknowledgement of authors' response**
> >
> > I have read in detail the responses of the authors to my comments and those of the other reviewers. I appreciate the clarifications provided and the additional experiences introduced, especially considering the short rebuttal period. However, I believe that a more systematic evaluation regarding the comparison with state-of-the-art algorithms would be beneficial (varying the length of inputs - which, if I understand correctly from the experiments, is 16, rather low for SOTA algorithms) and would better highlight the advantages of the proposed method. I understand the argument for the model-agnostic method, but if SOTA methods are close in terms of results to the authors' proposal, the utility of the method is not necessarily clear. Especially in real-world cases, where we do not know if the series contains anomalies or not: if the performance degrades in the absence of anomalies, the question then arises about the risk of using such a method. However, for the clarifications, experiments, and explanations provided, I am adjusting my rating to 5. Consolidated experiments seem necessary to further improve the rating (even though I understand that it may not be possible to conduct them in such a short time).

---

### Official Review · Reviewer_5W9f · 2023-11-03

**Soundness:** 3 good
**Presentation:** 4 excellent
**Contribution:** 3 good
**Rating:** 6
**Confidence:** 4

**Summary:**

This paper presents an algorithm for time series forecasting in the presence of anomalies, together with some theory and experimental results.

**Strengths:**

1. The paper explores a bit of variability in various hyper parameters and dataset normalization---more than the typical submission. Although, as I point out in weaknesses, some more information is needed.
2. The proofs are clearer than the typical ML paper.

**Weaknesses:**

1. The ranges of hyper parameters, noise scales, and stds, and other variables related to normalizing the data and running the algorithms need to be clearly identified.
2. In table 7, and some others where noise rates are varied, show that the lowest noise rate is 0.1. That is a rather high rate. The noise rate needs to be decreased to at least 0.01 and maybe lower depending on the dataset.
3. The characteristics of the chosen datasets need to be described. Hopefully they cover a large range of variability, relationships between input variables, fraction of dataset that have anomalies, etc.

Minor weakness:
1. "Proposation" -> "Proposition."

**Questions:**

1. In section 3, the fourth line, I suspect that $T-\alpha$ should be $T/\alpha$. Is that correct?
2. [Yoon 2022a] and [Yoon 2022b] are the same citation.
3. In appendix B, the lines after the first one seem to have been cut off and are hidden behind figure 2.
4. For table 12, what computer system was used for training?

---

> ### Author Response · Authors · 2023-11-21
> **Rebuttal-1**
>
> **W1:** The ranges of hyper parameters, noise scales, and stds, and other variables related to normalizing the data and running the algorithms need to be clearly identified.
>
> **A:** Thank you for your valuable suggestion. Regarding noise scales and standard deviations, these parameters are integral to our noise model, typically spanning the range of $(-\infty, +\infty)$. As detailed in Appendix E, our approach involves normalizing the training set to a mean of 0 and a standard deviation of 1, followed by the addition of anomalies. Specifically, for constant-type anomalies, the noise scale is $\epsilon = 0.5 \cdot \text{std} = 0.5$; for missing-type anomalies, the noise scale is $\epsilon = 0$; and for Gaussian-type anomalies, the noise scale is $\epsilon \sim \mathcal{N}(0,2\cdot \text{std})$. This choice is inspired by the initial paper on the TSFA task [R1]. In Appendix F.7, we illustrate how these noise modeling hyperparameters influence the performance of RobustTSF.
>
> Concerning the hyperparameters necessary for algorithm execution, the primary parameters are $\lambda$ and $\tau$, both with a range of $(0, +\infty)$. To demonstrate RobustTSF's robustness to hyperparameters, we consistently employ the same values across all settings and datasets in our experiments. In Appendix F.6, we provide insights into how the values of these algorithmic hyperparameters impact the performance of RobustTSF.
>
>
> [R1] Recurrent neural networks and robust time series prediction
>
> ----------------
>
> **W2:** In table 7, and some others where noise rates are varied, show that the lowest noise rate is 0.1. That is a rather high rate. The noise rate needs to be decreased to at least 0.01 and maybe lower depending on the dataset.
>
> **A:** We appreciate your insightful feedback. The choice of noise rates in our experiments aligns with conventional learning under noisy label settings [R1, R2] and relevant work on TSFA tasks [R3]. Very small noise ratios might not reveal significant differences among many methods, making it challenging to determine superiority.
>
> Acknowledging that real-world datasets may exhibit lower noise rates, contingent on the specific dataset, we conducted RobustTSF on datasets with real-world anomalies, as outlined in Appendix F.11. Additionally, we performed experiments with small noise ratios on the Electricity dataset and report the best epoch MAE, with results provided in the table below:
>
>
> |           | Const 0.01 | Const 0.05 | Missing 0.01 | Missing 0.05 | Gaussian 0.01 | Gaussian 0.05 |
> | --------- | ---------- | ---------- | ------------ | ------------ | ------------- | ------------- |
> | MAE       | 0.187      | 0.190      | 0.187        | 0.193        | 0.187         | 0.193         |
> | MSE       | 0.187      | 0.197      | 0.192        | 0.215        | 0.191         | 0.192         |
> | Offline   | 0.180      | 0.185      | 0.183        | 0.187        | 0.182         | 0.184         |
> | Online    | 0.187      | 0.193      | 0.188        | 0.190        | 0.188         | 0.190         |
> | Loss sel  | 0.192      | 0.195      | 0.192        | 0.196        | 0.187         | 0.190         |
> | RobustTSF | 0.177      | 0.177      | 0.178        | 0.179        | 0.177         | 0.177         |
>
> The results affirm that RobustTSF maintains superiority even under low noise ratios.
>
> [R1] Learning with Noisy Labels. NeurlPS 2013
>
> [R2] Peer Loss Functions: Learning from Noisy Labels without Knowing Noise Rates. ICML 2020
>
> [R3] Recurrent neural networks and robust time series prediction. TNNLS 1994

---

> ### Author Response · Authors · 2023-11-21
> **Rebuttal-2**
>
> **W3**: The characteristics of the chosen datasets need to be described. Hopefully they cover a large range of variability, relationships between input variables, fraction of dataset that have anomalies, etc.
>
> **A:**  We would like to clarify that, for the context of learning with noisy labels in time series forecasting with anomalies, datasets are typically selected through two main approaches. One involves manually introducing noise (anomalies) with defined noise ratios into a clean dataset, while the other uses datasets that naturally exhibit real-world anomalies.
>
> In our paper, we employed eight datasets, including Electricity, Traffic, ETT-h1, ETT-h2, ETT-m1, Exchange, with manually added anomalies, and NAB, M4 with real-world anomalies.
>
> Many of these datasets, such as Electricity and Traffic, are widely used in popular time series forecasting research. It's worth noting that even in datasets where anomalies are not manually added, they may still exist inherently due to environmental influences on sensors, such as temperature and humidity. Table 2 demonstrates the commendable performance of RobustTSF even in scenarios without manually added anomalies.
>
> In Appendix F.11, we further demonstrate the robust performance of RobustTSF on a dataset with real-world anomalies.
>
> We provide a comprehensive characterization of these datasets as follows:
>
> |     Dataset      |  LEN  |  FREQ   | Manullay adding anomalies | Train Test ratio |
> | :--------------: | :---: | :-----: | :-----------------------: | :--------------: |
> |   Electricity    | 26304 |   1h    |            Yes            |       7:3        |
> |     Traffic      | 17544 |   1h    |            Yes            |       7:3        |
> |      ETT-h1      | 17420 |   1h    |            Yes            |       7:3        |
> |      ETT-h2      | 17420 |   1h    |            Yes            |       7:3        |
> |      ETT-m1      | 69680 | 15 min  |            Yes            |       7:3        |
> |     Exchange     | 7588  |  1 day  |            Yes            |       7:3        |
> | NAB (real-cloud) | 4032  | 15 min  |            No             |       7:3        |
> |    M4-monthly    | 1871  | Monthly |            No             |       7:3        |
>
> ----------------------------------
>
> **W4:** Proposation" -> "Proposition."
>
> **A:** Thank you for pointing that out; we have corrected it accordingly.
>
> ----------------------------------
>
> **Q1:** In section 3, the fourth line, I suspect that $T - \alpha$ should be $T/\alpha$
>
> **A:** It should be $T - \alpha$ since the time series is partitioned into a set of time windows with size $\alpha$, and the total length of the time series is $T$.
>
> ----------------------------------
>
> **Q2:** [Yoon 2022a] and [Yoon 2022b] are the same citation.
>
>
> **A:** Thanks for bringing this to our attention. We have revised and consolidated the citation into one.
>
> ----------------------------------
> **Q3:** In appendix B, the lines after the first one seem to have been cut off and are hidden behind figure 2.
>
> **A:** Thanks for the careful examination. We have addressed and rectified this issue.
>
> ----------------------------------
>
> **Q4:** For table 12, what computer system was used for training?
>
> **A:** It was run on a MacBook with an Apple M1 chip. This information has been added to the paper.

---

### Author Response · Authors · 2023-11-21
**Response to all reviewers**

Dear AC and Reviewers,

We express our heartfelt appreciation to the AC and reviewers for their thorough evaluation of our paper and for recognizing our contributions in various dimensions:

- Novel and effective approach (R3, R4)
- Well-introduced theoretical aspects (R1, R2)
- Reproducible results (R2)
- Paving the way for new directions (R3)

The reviewers have primarily raised concerns regarding the evaluation of our experiments (R1, R3, R4), the comparative analysis with other methods (R2), the novelty of our approach (R2), and the clarity of some theoretical aspects (R3, R4). In response, we have meticulously addressed each concern with detailed explanations and supplementary experiments. Our paper has been revised accordingly to incorporate these changes, which are highlighted in blue.

We would appreciate it if the reviewers could provide feedback and adjust the rating accordingly after reading the rebuttal.

Best regards,
Authors

---

### Meta-Review · Area_Chair_Cgug · 2023-12-06

**Metareview:**

This paper considers more realistic settings for time series forecasting in which different types of anomalies exist in the training data. It aims to devise a robust forecasting model with theoretical justification. In response to the questions and concerns raised by the reviewers on the original submission, the authors replied them in detail and provided additional experiments to substantiate their claims. Two of the four reviewers also decided to increase their overall ratings. While this appears to be a borderline paper in terms of the overall ratings given by the reviewers, the novelty of the problem studied and the good balance between theoretical and empirical treatments of the paper provide strong arguments for accepting this work to the ICLR community. While even the revised version of the paper has room for improvement, the weaknesses do not imply rejection. The authors are highly recommended to further improve their work to address the outstanding concerns to make it appeal better to the research community.

**Justification For Why Not Higher Score:**

It is not good enough to be accepted as a spotlight.

**Justification For Why Not Lower Score:**

In my opinion, the overall ratings do not truly reflect the merits of this paper. Based on the author responses, one reviewer (442h) should have increased the overall rating from 5 to at least 6. As for another reviewer (FUy4) with a rating of 5, he made a remark on 1 Dec 2023 that “it’s not a problem for me if it is accepted”.

---

### Decision · Program_Chairs · 2024-01-16

Accept (poster)